# Revising chronological uncertainties in marine archives using global anthropogenic signals: *a case study on oceanic $^{13}$C Suess effect*

Nil Irvalı[1], Ulysses S. Ninnemann[1], Are Olsen[2], Neil L. Rose[3], David J. R. Thornalley[3], Tor L. Mjell[1,*], François Counillon[2,4]

[1]Department of Earth Science, University of Bergen and Bjerknes Centre for Climate Research, Bergen, 5007, Norway
[2]Geophysical Institute, University of Bergen and Bjerknes Centre for Climate Research, Bergen, 5007, Norway
[3]Department of Geography, University College London, London, WC1E 6BT, UK
[4]Nansen Environmental and Remote Sensing Centre and Bjerknes Centre for Climate Research, Bergen, 5007, Norway
*Now at: Slåtthaug School, Bergen kommune, Bergen, 5020, Norway

*Correspondence to*: Nil Irvalı (nil.irvali@uib.no)

**Abstract.** Marine sediments are excellent archives for reconstructing past changes in climate and ocean circulation. Overlapping with instrumental records they hold the potential to elucidate natural variability and contextualize current changes. Yet, dating uncertainties of traditional approaches (e.g., up to ±30-50 years, for the last two centuries) pose major challenges for integrating the shorter instrumental records with these extended marine archives. Hence, robust sediment chronologies are crucial and most existing age model constraints do not provide sufficient age control, particularly for the 20th century, which is the most critical period for comparing proxy records to historical changes. Here we propose a novel chronostratigraphic approach that uses anthropogenic signals such as the oceanic $^{13}$C Suess effect and spheroidal carbonaceous fly ash particles to reduce age model uncertainties in high-resolution marine archives. As a test, we apply this new approach to a marine sediment core located at the Gardar Drift, in the subpolar North Atlantic, and revise the previously published age model for this site. We further provide refined estimate of regional reservoir corrections and uncertainties for Gardar Drift.

## 1 Introduction

One of the most prominent features of 20th century climate in the circum-North Atlantic is the observed basin wide multi-decadal variations in the Atlantic Ocean Sea Surface Temperatures (SSTs)—the Atlantic Multidecadal Variability, AMV. This has impacts on the North American and European climate (Sutton and Hodson, 2005), frequency of Atlantic hurricanes (Goldenberg et al., 2001), extent of Arctic sea ice (Miles et al., 2014), as well as rainfall patterns in African Sahel (Wang et al., 2012). However, instrumental SST records are limited to the last ~150 years (e.g., Kaplan et al., 1998), and in only a few location – widespread coverage exist only since the 1950s onwards. Yet longer records of climate and ocean circulation are required to understand and assess the mechanisms behind its variability. For example, it is still debated whether AMV is driven internally, linked to multi-decadal variations in the Atlantic Meridional Overturning Circulation (AMOC) (Zhang et al., 2019), driven externally, e.g., due to solar and volcanic forcings (Otterå et al., 2010), or the timing of anthropogenic forcings (Booth

et al., 2012); or even such an oscillation exists at all (Mann et al., 2020). Annually-laminated mollusk shell archives offer the excellent chronological constraint required to investigate such questions, however they are limited to shelf locations and the range of proxies that can be applied in these archives is limited (Reynolds et al., 2016). Also overlapping with, and extending the instrumental records further back in time, marine sediments hold the potential to resolve these issues and contextualize current changes. New high resolution proxy records, particularly from the North Atlantic sedimentary drift sites are now emerging, closing the time gap between the modern and paleo-observations (e.g., Boessenkool et al., 2007; Mjell et al., 2016; Thornalley et al., 2018; Spooner et al., 2020). For instance, Mjell et al. (2016) found that AMV and deep ocean circulation varied on similar timescales over the last 600 years, however, due to age model uncertainties as high as the duration of half an AMV cycle, determining the precise phasing was not possible and required independent age constraints. Hence, integrating near continuous but shorter observational records to longer (but with relatively lower resolution) marine archives still poses as one of the major challenges for the (paleo)oceanographic community.

Recent marine sediments are dated using an array of approaches, all of which have their own limitations and uncertainties. Radiocarbon ($^{14}$C) dating is one of the most common methods for dating marine sediment cores. The uncertainties with this method can exceed 50 years and include several caveats and assumptions such as uncertain and variable reservoir effects and confounding influences such as the effect of fossil fuel emissions on atmospheric radiocarbon and the H-bomb $^{14}$C spike, which further increases the uncertainties when dating recent sediments (Reimer et al., 2004; Hughen, 2007; Graven, 2015). In the latter case, the $^{14}$C bomb spike can provide as an additional high-resolution dating tool in marine settings, yet, this requires annually resolved archives (Scourse et al., 2012). Geochemical composition of tephra shards and fingerprinting these to known volcanic eruptions can also provide absolute age markers. The precision of these age markers can be 1-2 years, yet this method is only regionally applicable and the occurrence of multiple, closely spaced eruptions with similar geochemistry can lead to greater uncertainty (Lowe, 2011). A combination of radionuclide dating ($^{210}$Pb, $^{137}$Cs, $^{241}$Am) (Appleby, 2008) and more recently the increases in mercury (Hg) concentrations (i.e., as an anthropogenic (pollution) indicator) are used as chronostratigraphic markers on recent marine sediments (Moros et al., 2017; Perner et al., 2019). For instance, $^{210}$Pb dating is widely used for dating recent sediments (0-150 years), while chronostratigraphic markers such as the nuclear weapons test fallout in 1963 and Chernobyl fallout in 1986 can also be determined from the presence of $^{137}$Cs (Appleby, 2008). Still, $^{210}$Pb-based age models also involve multiple assumptions and are ideally validated using an independent age marker (e.g., $^{137}$Cs or $^{241}$Am) to assess the influence of post depositional remobilization or bioturbation. Yet, it remains difficult to confirm to what extent the assumptions for dating are met (Smith, 2001). $^{137}$Cs profiles are often used to partially validate $^{210}$Pb chronologies, but this can only be undertaken at specific periods (e.g. bomb-testing, Chernobyl). In addition, $^{137}$Cs is also prone to post depositional remobilization, and is not always above the detection limit—depending on core locations (e.g., Barsanti et al., 2020). Although the application of $^{210}$Pb dating in combination with $^{137}$Cs in lacustrine environments is well established, delayed input from $^{137}$Cs fallouts highlights the need for care in using $^{137}$Cs as chronostratigraphic markers even in lake sediments (Appleby et al., 2023). The situation is considerably more difficult in marine environments (e.g., Appleby et al.,

2021). Indeed, a recent review highlights the continuing importance of, and need for, independent age control markers to corroborate [210]Pb-based age models (Barsanti et al., 2020). Clearly progress is needed to improve age constraints in the 20th century in a way that will allow us to calibrate proxies using observational timeseries and, ultimately, reliably extend these observational records. Anthropogenic signals, such as the oceanic [13]C Suess effect and spheroidal carbonaceous fly ash particles (SCPs), are evident in high-resolution marine archives, and hold the potential to provide a means for improving age control over the 20th century.

Atmospheric $CO_2$ has been increasing due to human activities, such as fossil fuel combustion and deforestation, since the beginning of the industrial period. Due to preferential uptake of the lighter isotope (i.e., [12]C), increased anthropogenic $CO_2$ emissions cause the [13]C/[12]C ratio ($\delta^{13}C$) and the [14]C/C ratio ($\Delta^{14}C$) to decline. The decreasing trend in the radiocarbon ([14]C/C) content of $CO_2$ was first named as the "Suess effect" by (Suess, 1955). In 1979, due to its similarity, (Keeling, 1979) has extended the Suess effect terminology to the shifts in the [13]C/[12]C ratio of the atmospheric $CO_2$. The [13]C Suess effect propagates into different reservoirs of the Earth system, for instance, the addition of low $\delta^{13}C$ anthropogenic $CO_2$ from the atmosphere into the surface ocean also affects the natural $\delta^{13}C$ gradients (Eide et al., 2017; Olsen and Ninnemann, 2010). Foraminiferal $\delta^{13}C$ records (planktonic and benthic) from high-resolution marine archives capture this accelerating decline in $\delta^{13}C$ over the last century (e.g., Mellon et al., 2019), and thus hold a huge potential for refining age control for recent sediments.

Another new and promising approach for dating recent marine sediments is the use of spheroidal carbonaceous fly ash particles (SCPs)(Spooner et al., 2020; Thornalley et al., 2018). SCPs are only produced from the high temperature industrial sources, such as coal and oil, and thus are purely anthropogenic in origin. They are emitted to the atmosphere along with combustion flue gases and are therefore transported to and recorded in many natural archives worldwide— including regions that are remote from industrial sources (e.g., Rose et al., 2004; Rose et al., 2012). In lake sediment records, SCPs were first observed during the mid-19th century in the UK, Europe and North America, and show a very distinct concentration profile. The SCP concentration trend starts with a gradual increase from the beginning of the SCP record until the mid-20th century, followed by a rapid increase at c. 1950 linked with the increased demand for electricity following the Second World War (Rose, 2015). The beginning of the SCP record may vary regionally because it depends on the regional developments in industrial history as well as the sedimentation rates. However, the rapid increase observed in the mid-20th century has been considered to be a global signal (Rose, 2015) – making SCPs a robust and ideal stratigraphic marker for a mid-20th century Anthropocene. First applications of the SCP method to marine sediment archives (Thornalley et al., 2018; Spooner et al., 2020; Kaiser et al., 2023) have shown to follow the similar trends to those established from the lake records (Rose, 2015), providing an independent means to improve marine based chronologies over the last 150 years.

Here we combine these two novel chronostratigraphic approaches that use anthropogenic signals (i.e., oceanic [13]C Suess effect change and spheroidal carbonaceous fly ash particles (SCPs)) to reduce age model uncertainties in high-resolution marine

archives. As a test, we apply this new approach to a high-resolution site at the Gardar Drift, off southern Iceland to revise the previously published age model at this site (i.e., Mjell et al., 2016). We further provide refined regional $^{14}$C reservoir corrections and uncertainties for Gardar Drift, using a combination of $^{14}$C AMS dates and oceanic $^{13}$C Suess effect estimates for our core location.

## 2 Material and Methods

In this study we use sediment samples from the Gardar Drift Multicore, GS06-144-09 MC (60°19 N, 23°58 W, 2081 m water depth) recovered during the University of Bergen Cruise No: GS06-144, onboard the research vessel R/V *G.O. Sars*. Four successful identical cores (GS06-144-09 MC A-D) were recovered at this station. The 44.5 cm long GS06-144-09 MC-D has been sampled at 0.5 cm intervals. Each sample was soaked in distilled water and shaken for 12 hours in order to disperse the sediment, before they were wet-sieved and separated into size fractions of >63-µm and <63-µm. The fine fractions (<63 µm) were used for mean sortable silt grain size analysis (Mjell et al., 2016), whereas the >63-µm fraction was used for selection of foraminifera for stable isotope analysis  and $^{14}$C AMS dating (Table 1). The 44 cm long GS06-144-09 MC-C was  sampled at 0.5 cm intervals. Each sample was dried and weighed. Dry bulk sediment samples from the GS06-144-09 MC-C were used for SCP analysis.

Samples from GS06-144-09 MC-D have previously been analyzed for the activity of $^{210}$Pb, $^{226}$Ra and $^{137}$Cs at the Gamma Dating Centre, Department of Geosciences and Natural Resource Management, University of Copenhagen, Denmark (Mjell et al., 2016). The initial age model of GS06-144-09 MC-D was based on $^{210}$Pb excess dates from the top 7.25 cm and two $^{14}$C AMS dates (Mjell et al., 2016). Presence of $^{137}$Cs in marine sediment cores is often used to validate the $^{210}$Pb chronologies and can also provide additional information (e.g., an independent tie point) for the onset of atmospheric weapon testing (e.g., Perner et al., 2018). In Core GS06-144-09 MC-D, the content of $^{137}$Cs was very low and below detection limit except for the top 4 cm of the core. This may indicate that the top 4 cm could be younger than ~1950 AD. However, in the case for Core GS06-144-09 MC-D, traces (near detection limit) of $^{137}$Cs was also episodically present below this depth (Supplementary Figure S1 in Mjell et al., 2016). Hence, here we choose not to include the information provided by $^{137}$Cs in our age model, and neither include the $^{210}$Pb dates, as it will not be possible to validate with $^{137}$Cs.

In general, an ideal approach to build the best possible chronology is to integrate all available information. However, here we aim to demonstrate the potential utility of two novel approaches, oceanic $^{13}$C Suess effect change and SCPs, in building robust marine sediment chronologies. Therefore, we focus on these two novel techniques in a more standalone manner to assess their utility independently and their consistency with each other. Our methods include stable carbon isotopes of planktonic foraminifera ($\delta^{13}$C), $^{14}$C AMS dates, SCP analysis and time series of oceanic $^{13}$C Suess effect change computed for our core location.

**Table 1.** [14]C AMS Dates from GS06-144-09 MC-D.

| Lab Code | Depth (cm) | Material | [14]C age | ±1σ | Reference |
|----------|-----------|----------|-----------|-----|-----------|
| KIA 34242 | 0 | *N. incompta* | 75 | 20 | Mjell et al., 2016 |
| BE-19497.1.1 | 2.5 | *N. incompta* | 526 | 29 | This study |
| BE-19498.1.1 | 4 | *N. incompta* | 565 | 29 | This study |
| BE-19499.av | 5.5 | *N. incompta* | 603 | 48 | This study |
| BE-19500.av | 8 | *N. incompta* | 587 | 73 | This study |
| BE-19501.1.1 | 10 | *N. incompta* | 604 | 29 | This study |
| KIA 34243 | 11.5 | *N. incompta* | 530 | 20 | Mjell et al., 2016 |
| BE-19502.1.1 | 17.5 | *N. incompta* | 664 | 29 | This study |
| BE-19503.1.1 | 25.5 | *N. incompta* | 817 | 40 | This study |
| KIA 34244 | 30 | *N. incompta* | 750 | 20 | Mjell et al., 2016 |
| BE-19504.1.1 | 43 | *N. incompta* | 1226 | 30 | This study |


## 2.1 Stable isotope analysis ($\delta^{13}$C)

Stable isotope analyses ($\delta^{13}$C) were performed on planktonic foraminifera *Globigerina bulloides*, *Neogloboquadrina incompta* and *Globorotalia inflata* at every 0.5 cm resolution throughout the core . *G. bulloides* was picked from the 250-300 µm size fraction, while *N. incompta* was picked from the 150-250 µm and *G. inflata* was picked from the 250-350 µm size fractions.

Approximately 5-7 shells of *G. bulloides*, ~5 shells of *G. inflata* and ~10 shells of *N. incompta* from each sample were used for stable isotope analysis. Foraminifera were ultrasonically rinsed for 20 seconds in methanol to remove any contaminants prior to analysis. Stable isotope analyses were measured using a Finnigan MAT 251 and a MAT 253 mass spectrometer at the FARLAB (Facility for Advanced Isotopic Research), at the Department of Earth Science, University of Bergen. All samples were run in two replicates whenever foraminifera were sufficiently abundant. The stable isotope results are expressed as the

average of the two replicate measurements and reported relative to Vienna Pee Dee Belemnite (VPDB), calibrated using NBS-19. Long-term analytical precision (1σ) of the standards over the analysis period was better than 0.04‰ for $\delta^{13}$C.

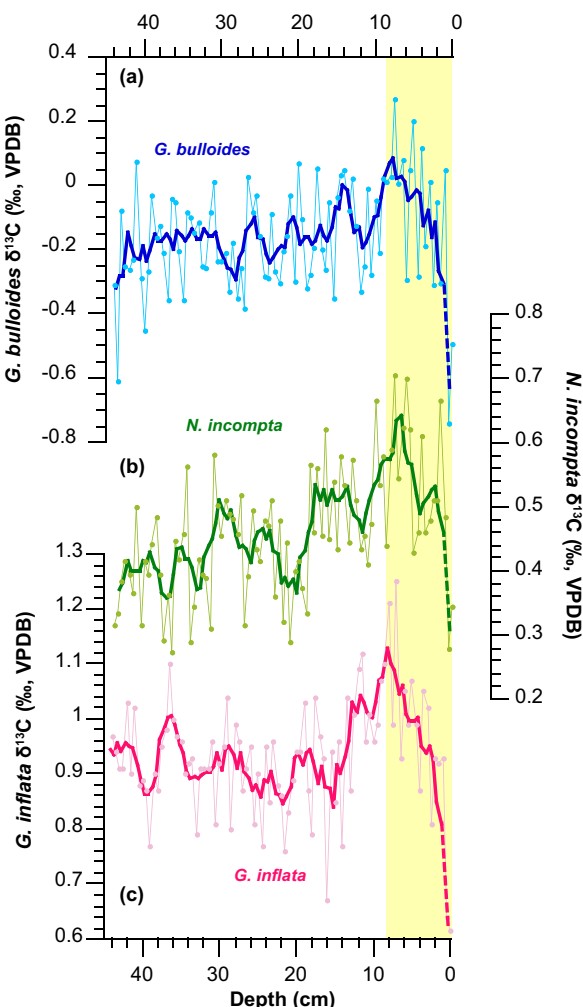

**Figure 1.** Planktonic $\delta^{13}$C records from Site GS06-144-09 MC-D plotted vs depth (cm). Yellow highlight marks the sharp decline in $\delta^{13}$C due to Suess effect. (a) *G. bulloides* $\delta^{13}$C record (blue) with 5-point mean (bold line), (b) *N. incompta* $\delta^{13}$C record (green) with 5-point mean (bold line), (c) *G. inflata* $\delta^{13}$C record (pink) with 5-point mean (bold line). The 5-point mean is extended into the core top, by taking the mean of samples at 0 and 0.5 cm; dashed bold lines, to highlight the large abrupt $\delta^{13}$C decrease at the core top.

## 2.2 $^{13}$C Suess effect estimates

Recently, Eide et al. (2017) calculated globally gridded surface to seabed $^{13}$C Suess effect estimates for the industrialized era. These estimates were based on the two-step back calculation technique of Olsen and Ninnemann (2010) for waters deeper than 200 m, while for waters above they were determined by combining the 200 m level estimate with values of the surface ocean $^{13}$C Suess effect as evident in coral and sclerosponge records. The two-step back calculation approach first takes advantage of the relationships between preformed $\delta^{13}$C and chlorofluorocarbons (CFC-11 or CFC-12) in the ocean to quantify the $^{13}$C Suess

effect since CFCs first appeared in the atmosphere (the 1940s). In the second step, these estimates are extended to the full

industrialized era under the assumption of Transient Steady State (Gammon et al., 1982; Tanhua et al., 2007), which states that after an initial adjustment period, the response in tracer concentrations at depth will be proportional to the change in boundary concentration in exponentially forced systems. This means that we can expect that the ratio of the [13]C Suess effect at any point in the ocean to that in the atmosphere will remain constant in time, i.e.:


$$\frac{\delta^{13}C_{SE,\Delta t1}^{ocean}}{\delta^{13}C_{SE,\Delta t1}^{atm}} = \frac{\delta^{13}C_{SE,\Delta t2}^{ocean}}{\delta^{13}C_{SE,\Delta t2}^{atm}}$$ (Eq. 1)

where $\Delta t1$ and $\Delta t2$ represents two time intervals since the preindustrial. In the case of Eide et al. (2017) these are the periods 1940 to 1994 and preindustrial (defined as atmospheric $\delta^{13}C$=-6.5) to 1994.


Here, we use Eq. (1) to derive time series of the Suess effect since the preindustrial at 10 depth layers from the surface to 200 m (e.g, $\delta^{13}C_{SE\_0}$, $\delta^{13}C_{SE\_50}$), above the Gardar Drift core site. This depth interval covers the depth habitats of the planktonic foraminiferal species we have used for stable isotope analysis. The time series were determined by taking the ratio between the Suess effect determined by Eide et al. (2017) at each of the 10 depth levels we consider in the grid box covering the Gardar

drift (60°-61°N and 23°-24°W) and the atmospheric $\delta^{13}C$ decline until 1994 and multiplying this with the atmospheric $\delta^{13}C$ history since preindustrial provided by Rubino et al. (2013). The thus calculated marine Suess effect time series are presented in Fig 2. We set the starting point in time to 1800, as an appreciable decline in atmospheric $\delta^{13}C$ is only visible after that year.

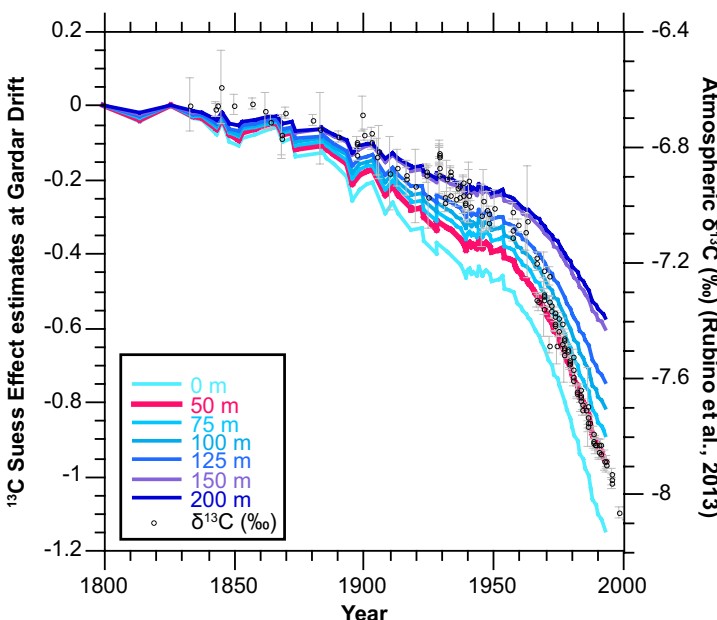

**Figure 2.** [13]C Suess effect estimates at the Gardar Drift (60.5°N, 23.5°W), for the 10 different depth layers from the surface to 200 m, plotted together with the atmospheric $\delta^{13}C$ record provided by Rubino et al. (2013).

## 2.3 SCP analysis

We followed the SCP method outlined by Rose (1994). Approximately 0.2 g of dried bulk sediment was weighed into 15 ml polypropylene tubes. One SCP reference standard (Rose, 2008) and a blank were included for quality control purposes and treated exactly the same as the samples. The SCP extraction method included nitric acid ($HNO_3$), hydrofluoric acid (HF) and hydrochloric acid (HCl) stages to respectively remove organic matter, silicious material and carbonates. Following the acid digestion stages, a known fraction of the final residue was evaporated onto a cover slip and mounted on a microscope slide using Naphrax mountant. A light microscope with 400x magnification was used to identify and count the total number of SCPs on each slide. SCP identification followed the criteria described in Rose (2008) based on morphology, color, depth and porosity. SCP concentrations are reported as number of SCPs per gram dry sediment (gDM[-1]). SCP analyses were performed at the Department of Geography, University College London. The concentration of the SCP reference material was 5318 gDM[-1] (±1022, 90% confidence level), close to the reported concentration of 6005±70 gDM[-1] (Rose, 2008). No SCPs were observed in the blank.

## 3 The new age model approach

### 3.1 Planktonic foraminiferal $\delta^{13}C$ vs oceanic $^{13}C$ Suess effect

In the subpolar North Atlantic, *G. bulloides* calcifies in the upper 50 m of the water column, over the late spring and summer, depending on food availability (Jonkers et al., 2013; Schiebel et al., 1997; Spero and Lea, 1996; Chapman, 2010). On the other hand, the habitat depth of *N. incompta* is highly variable, ranging from surface to deeper thermocline, most likely calcifying between 50 to 125 m water depth (Chapman, 2010; Field, 2004; Pak and Kennett, 2002; Pak et al., 2004; Von Langen et al., 2005; Nyland et al., 2006; Schiebel et al., 2001). *G. inflata* is a deep dwelling foraminiferal species, living at the base of the seasonal thermocline or deeper in the main thermocline if the base of the seasonal thermocline is warmer than 16°C (Cléroux et al., 2007). In the North Atlantic, *G. inflata* calcifies between 200 and 400 m south of 57°N, and between 100 and 200 m north of 57°N (Ganssen and Kroon, 2000).

To calculate the age estimates based on $^{13}C$ Suess effect, we assume a calcification depth of 50 m for *G. bulloides* and compare our *G. bulloides* $\delta^{13}C$ record with the $^{13}C$ Suess effect change at 50 m ($\delta^{13}C_{SE\_50}$), at our core location. In order to avoid any uncertainties regarding planktonic foraminiferal depth habitats, we also present a stacked planktonic $\delta^{13}C$ record ($\delta^{13}C_{stack}$; i.e., the average of *G. bulloides*, *N. incompta* and *G. inflata*) and compare it with the average $^{13}C$ Suess effect change over the top 200 m of the water column ($\delta^{13}C_{SE\_0-200}$), which spans the depth habitats of all three planktonic species (i.e., *G. bulloides*, *N. incompta* and *G. inflata*) used in this study (Supplementary Figures 1 and 2).

### 3.1.1 *G. bulloides* $\delta^{13}C$ vs oceanic Suess effect change at 50 m ($\delta^{13}C_{SE\_50}$)

*G. bulloides* $\delta^{13}C$ record shows large natural variability over the 10-44 cm core interval, varying between ~0.08 ‰ and ~-0.6 ‰. However, the most prominent feature occurs towards the core top. $\delta^{13}C$ values reach a peak of 0.27 ‰ at 7.5 cm, start to gradually decrease and reach 0.05 ‰ at 1 cm. This is followed by a very sharp decline of ~0.8 ‰ centered at 0.5 cm. The gradual decrease observed in *G. bulloides* $\delta^{13}C$—with a sharper decline at the core top indicates the presence of the $^{13}C$ Suess effect. Compared to the $^{13}C$ Suess effect change at 50 m, the relative change in *G. bulloides* $\delta^{13}C$ seems to be very similar (Supplementary Figure 3). Does the $\delta^{13}C_{SE\_50}$ curve provide a means to narrow down chronological uncertainties over the industrial period? To explore this, we objectively matched our *G. bulloides* $\delta^{13}C$ record with the $\delta^{13}C_{SE\_50}$ curve to find the starting point (1800 AD) of the Suess effect curve on the *G. bulloides* $\delta^{13}C$ record.

To objectively place the start of the $\delta^{13}C_{SE\_50}$ curve (1800 AD) on the *G. bulloides* $\delta^{13}C$ record, first we computed the curvature of the $\delta^{13}C_{SE\_50}$ curve. We use a 3rd degree polynomial fit, using the polyfit function in MATLAB. Secondly, we apply 3rd degree polynomial curve fits to the *G. bulloides* $\delta^{13}C$ record, for different core depth intervals (n=12), starting from 12 cm to cover the whole industrial period. We apply curve fits to 12-0 cm, 11-0 cm, 10-0 cm, 9-0 cm, 8.5-0 cm, 8-0 cm, 7.5-0 cm, 7-0

cm, 6.5-0 cm 6-0 cm, 5.5-0 cm and 5-0 cm intervals. When applying curve fits, we use *G. bulloides* $\delta^{13}$C, as well as its 3-point running mean and 5-point running mean, assuming the overall trends might be better represented in the smoothed data. Goodness of fit results for each curve fit is presented in Supplementary Table 1. Finally, we compared the curvature of the $\delta^{13}C_{SE\_50}$ curve with the various curve fits applied to *G. bulloides* $\delta^{13}$C records and find which curve fit is the most similar to the curvature of the $\delta^{13}C_{SE\_50}$. To do this, we calculate the correlation coefficients between our target curve (in this case, the curvature of $\delta^{13}C_{SE\_50}$) and each of the $3^{rd}$ degree polynomial curves, using their individual polynomial coefficients (i.e., p1, p2, p3, p4; Supplementary Table 1). The curvature of the *G. bulloides* $\delta^{13}$C record for the 7.5-0 cm interval is the most similar to curvature of our $\delta^{13}C_{SE\_50}$ curve (r = 0.73), suggesting 7.5 cm could be 1800 AD. We further do the same test using the 3-point and 5-point running mean of the data. Although the correlation is poorer, the same result is also reached (i.e., best fit when record starts at 7.5 cm) when 3-point running mean (r = 0.46) and 5-point running mean (r = 0.22) of *G. bulloides* $\delta^{13}$C is used. Placing the start of the oceanic Suess effect change (~1800 AD) on our *G. bulloides* $\delta^{13}$C record is one of the main challenges of our approach as the most prominent $\delta^{13}$C decline does not happen until the most recent years/core top. This is also evident from our correlation analysis results (Supplementary Table 1). For instance, a close second-best fit occurs when we place 1800 AD at 5 cm (r = 0.69) instead of 7.5 cm (r = 0.73). A comparison between the two correlation coefficients using the Fisher's z transformation suggests that the difference between the correlation coefficients is not statistically significant (z = 0.064, p = 0.949). This indicates that 5 cm could also be 1800 AD.

### 3.1.2 $\delta^{13}C_{stack}$ vs the 0-200 m average of oceanic suess effect change ($\delta^{13}C_{SE\_0-200}$)

*N. incompta* and *G. inflata* $\delta^{13}$C also followed a very similar trend as the *G. bulloides* $\delta^{13}$C record – with the most prominent decline towards the core top, indicating the presence of the $^{13}$C Suess effect. To cross-check our approach described in 3.1.1 and to avoid any uncertainties that may be caused due to habitat depth variability, we use the stacked planktonic $\delta^{13}$C of *G. bulloides*, *N. incompta* and *G. inflata* ($\delta^{13}C_{stack}$). Considering the habitat depth range of all three planktonic species, we then compare the $\delta^{13}C_{stack}$ with the 0-200 m average of the $^{13}$C Suess effect ($\delta^{13}C_{SE\_0-200}$) (Supplementary Figures 1 and 2). Similarly, to place the start of the $\delta^{13}C_{SE\_0-200}$ curve (1800 AD) on our $\delta^{13}C_{stack}$ record, first we find the curvature of the $\delta^{13}C_{SE\_0-200}$ curve. We use a $3^{rd}$ degree polynomial fit, using the polyfit function in MATLAB. Secondly, we apply $3^{rd}$ degree polynomial curve fits to the $\delta^{13}C_{stack}$ record, for the same core depth intervals as in 3.1.2. Finally, we compare the curvature of the $\Delta^{13}C_{SE\_0-200}$ curve with the various curve fits applied to our $\delta^{13}C_{stack}$ record and find which curve fit is the most similar to the curvature of $\delta^{13}C_{SE\_0-200}$. To do this, we again calculate the correlation coefficients between our target curve (in this case, the curvature of $\delta^{13}C_{SE\_0-200}$) and each of the $3^{rd}$ degree polynomial curves, using their individual polynomial coefficients (i.e., p1, p2, p3, p4; Supplementary Table 1). In this case, we get similar results for intervals 5-0 cm, 5.5-0 cm and the 7.5–0 cm (r = -0.60). Although, the negative correlation coefficients indicates that the similarity approach used here may not capture the complexity

of comparing 3$^{rd}$ degree polynomials, it gives us a rough estimate of which curve is most similar to our target curve (i.e., $\delta^{13}C_{SE\_0\text{-}200}$), and overall agrees with our initial finding based on *G. bulloides* that 7.5 cm or 5 cm may in fact be 1800 AD.

### 3.2 Core top age

In paleoceanographic studies it is common to use the year a sediment core was retrieved as the core top age. However, this is highly dependent on the sedimentation rates of the region and may not always be the case. The core top (0 cm) $^{14}C$ AMS date for GS06-144-09 MC-D indicated the presence of bomb carbon, confirming that the top should be younger than ~1957 AD (Mjell et al., 2016). Therefore, based on high sedimentation rates at the site Mjell et al. (2016) assumed 2006 AD as the core top age, i.e., the year Core GS06-144-09 MC was retrieved. Here we explore this further considering the new information provided by the relative change in our oceanic $^{13}C$ Suess effect curve. For this, we use the *G. bulloides* $\delta^{13}C$ record and the $\delta^{13}C_{SE\_50}$ curve.

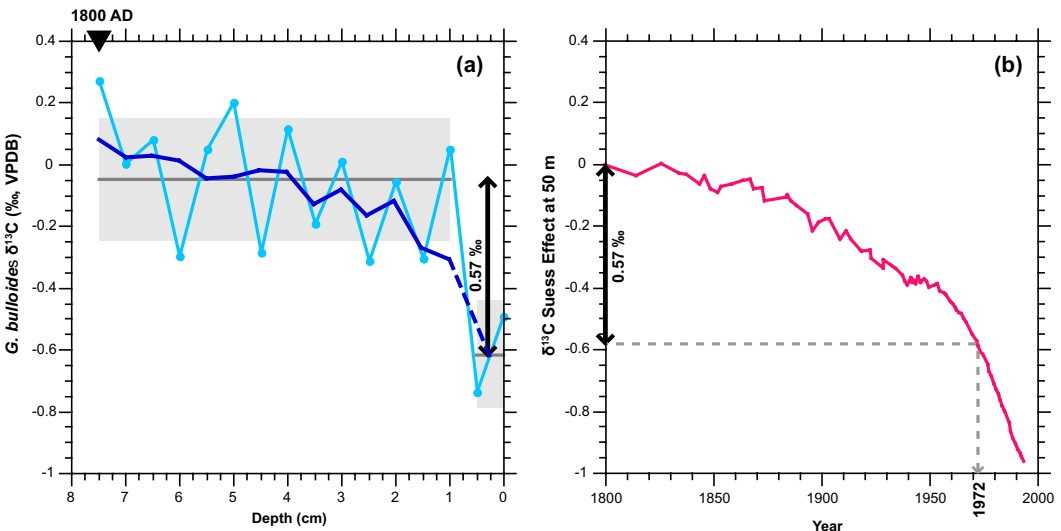

**Figure 3.** Overview of core top age calculation. (a) *G. bulloides* $\delta^{13}C$ record (blue, with 5-point mean (bold line)) vs depth (cm). The 5-point mean is extended into the core top, by taking the mean of samples at 0 and 0.5 cm; dashed bold lines, to highlight the large abrupt $\delta^{13}C$ decrease at the core top. Dark gray line and gray shadings respectively mark the mean and standard deviation of the 1-7.5 cm and 0 and 0.5 cm intervals. (b) $\delta^{13}C_{SE\_50}$ curve (pink). Arrow and dashed lines mark when a 0.57 ‰ magnitude decline occurs in the record.

Based on our previous curve fits, we place 1800 AD at 7.5 cm. The most prominent change in *G. bulloides* $\delta^{13}C$ record occurs at 0.5 cm. Hence, first we find the mean and standard deviation of the 1-7.5 cm interval (-0.05 ± 0.2; n=14), i.e., the mean $\delta^{13}C$ over the industrial period, and secondly the core top (two data points at 0 and 0.5 cm; -0.62 ± 0.17; n=2) of the *G. bulloides* $\delta^{13}C$ record, i.e., where the sharpest decline due to Suess effect occurs. We then calculated the difference in means using a t-test (0.57 ‰) and found the magnitude of the sharpest decline in *G. bulloides* $\delta^{13}C$ due to Suess effect. Finally, we used the

$\delta^{13}C_{SE\_50}$ curve to find when a 0.57 ‰ magnitude decline relative to the preindustrial value occurred. Based on our $\delta^{13}C_{SE\_50}$ curve, a decline of 0.57 ‰ occurs in ~1972. This would then place 1972 AD at 0.25 cm (i.e., the mid-point of our two samples at 0 and 0.5 cm), suggesting a much older core top age than previously assumed for GS06-144-09 MC (Mjell et al., 2016). We repeated the same approach and evaluated how the core top age would change if we placed 1800 AD at 5 cm (i.e., our second best-fit). We computed the mean and standard deviation of the 1-5 cm interval (-0.09 ± 0.2; n=9) and calculated the difference in means (0.53 ‰) between the 1-5 cm interval and the core top (0-0.5 cm). Finally, we determined when a magnitude of 0.53 ‰ decline occurred in the $\delta^{13}C_{SE\_50}$ curve. Based on our $\delta^{13}C_{SE\_50}$ curve, a decline of 0.53 ‰ occurred in ~1969, placing 1969 AD at 0.25 cm. This suggests that placing 1800 AD at 7.5 cm vs 5 cm changes our core top age (or our tie point at 0.25 cm) by 3 years. When building our age model, here we choose 7.5 cm as 1800 AD (i.e., based on the best curve fit) and 0.25 cm as 1972 AD, and introduced a 3 years' uncertainty on the selection of these tie points.

### 3.3 Revising regional reservoir corrections (ΔR) at Gardar Drift

To build an age model for the marine sediment cores based on radiocarbon dating it is necessary to convert $^{14}C$ dates into calendar years. Surface ocean $^{14}C$ is depleted relative to the atmosphere, which is known as the marine reservoir effect. Global marine radiocarbon calibration curves, e.g., the latest Marine20 Curve (Heaton et al., 2020), account for the global average offset between the marine and atmospheric reservoirs, however, there are temporal and spatial deviations from this offset. Marine reservoir ages range from 400 years in subtropics to more than 1000 years in polar oceans (Key et al., 2004). Therefore, the accurate calibration of $^{14}C$ ages depends on the knowledge of the local radiocarbon reservoir age of the surface ocean, i.e. the regional difference (ΔR) from the global marine radiocarbon calibration curve. The marine reservoir database within CALIB (http://calib.org/marine/) is the most extensive and valuable source for ΔR values for the modern ocean (Reimer and Reimer, 2001; Stuiver and Reimer, 1986). This online platform provides the user with an average ΔR value for their core location, based on the information provided on coordinates and number of nearest points. The ΔR values within the marine reservoir database are determined based on the known-age approach, i.e., when the death date (in calendar ages) of a pre-bomb marine sample (e.g., a mollusk shell) is known. However as a consequence of nuclear tests in the 1950s and early 1960s the ΔR calculation with the known-age approach can only be applied to samples collected before 1950 AD, hence the majority of the samples within the marine reservoir database are not homogenously distributed – making it temporally and spatially limited (Alves et al., 2018). Therefore, deriving a ΔR using the nearest points to a core location is problematic for many regions, where a closest ΔR is either not available or located at a different oceanographic setting (e.g., Hinojosa et al., 2015). When selecting samples for ΔR calculation, it is also important to review the ecological information on the taxa which the ΔR value is derived from, as some studies find species specific values due to habitat, feeding mechanisms and food sources. For instance, suspension feeders are thought to be the most suitable for dating, whereas deposit feeders, omnivore or carnivorous species are generally excluded due to their greater uncertainty in $^{14}C$ ages as they incorporate old carbon (Pieńkowski et al., 2021; England et al., 2013; Forman and Polyak, 1997). However, some studies find no difference in $^{14}C$ ages due to feeding mechanisms when the mollusks are derived from areas with no carboniferous rocks or local freshwater inputs to surface ocean

(Ascough et al., 2005).

Supplementary Table 2 shows the ΔR values for our core site (GS06-144-09 MC; 60°19 N, 23°58 W), located south of Iceland, derived from the nearest points available in the marine reservoir database (Reimer and Reimer, 2001). When the 10 nearest points are used (i.e., based on the distance (km) from core location), the ΔR for our core site is -72±64 [14]C yr. However, when we exclude carnivore and deposit feeding species, the ΔR value becomes -80±54 [14]C yr. It is also important to note that even the individual samples have a large range of ΔR values, varying between -23±45 to -220±85 [14]C yr, suggesting there might be other factors influencing the ΔR. For instance, considering the oceanographic setting, another approach could be to only select samples located around southern Iceland– i.e., those potentially under the influence of the Irminger Current, where our core site lies. Then, the ΔR value would be -92±93 [14]C yr (or -126±66 [14]C yr when carnivore and deposit feeding species are excluded). This suggests the available ΔR values within the CALIB marine reservoir database (Reimer and Reimer, 2001) for the region is highly variable and highly dependent on the selection criteria used by the investigator.

Global Ocean Data Analysis Project (GLODAP) radiocarbon observations (Key et al., 2004) provides an alternative approach to estimate the spatial variations in the reservoir ages (Gebbie and Huybers, 2012; Waelbroeck et al., 2019). For instance, Waelbroeck et al. (2019) have extracted the pre-bomb surface mean (upper 250 m) reservoir ages from the re-gridded (4° x 4°) GLODAP data. Following Waelbroeck et al. (2019)'s approach, we extract the reservoir ages (443 ± 75.8 [14]C yr) at our core site (60°N, 24°W) from the GLODAP data. Waelbroeck et al. (2019) note that, however, the error for their reservoir ages should be at least 100 [14]C yr, if the computed GLODAP standard deviation is less than this value (i.e., in our case: 443 ± 100 [14]C yr). Considering the global average marine reservoir age of ~600 years based on the Marine20 (Heaton et al., 2020), this would suggest a ΔR of -157±100 [14]C yr for our region. The large difference (and/or uncertainties) in regional reservoir corrections extracted using two independent methods (e.g., CALIB vs GLODAP-based) highlights the need for additional approaches to further constrain regional reservoir ages.

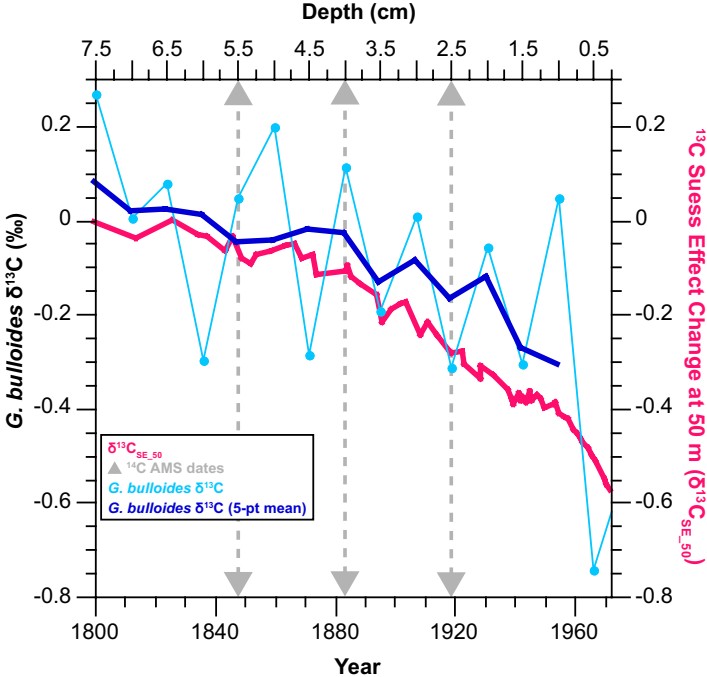

**Figure 4.** *G. bulloides* $\delta^{13}$C (7.5 – 0.25 cm (light blue line plotted with 5-pt mean (dark blue bold line)) vs the $\delta^{13}C_{SE\_50}$ curve spanning the 1800-1972 AD interval (pink bold line). Gray triangles on the depth axis mark the three $^{14}$C AMS samples at 2.5 cm, 4 cm and 5 cm depth intervals, while the gray dashed lines and triangles on the age axis mark their corresponding "known-ages" based on the $\Delta^{13}C_{SE\_50}$ comparison.

Here we suggest an alternative approach for calculating the $\Delta$R for marine sediment cores, that is independent of uncertainties such as the distance between core sites and sample locations, different oceanographic settings (e.g., coastal/fjord regions vs open ocean) or feeding ecology of the species used for dating. Based on our comparison of *G. bulloides* $\delta^{13}$C record and the $\delta^{13}C_{SE\_50}$ curve we obtain two tie points, placing 1972 AD at 0.25 cm and 1800 AD at 7.5 cm. Figure 4 shows the *G. bulloides*

$\delta^{13}$C record on depth scale spanning the 7.5 - 0.25 cm core interval, plotted together with the $\delta^{13}C_{SE\_50}$ curve spanning the 1800-1972 AD interval. First, we estimate the "known ages" for depths 2.5 cm, 4 cm and 5.5 cm (i.e., where we have $^{14}$C dates) by reading the corresponding ages from the $\delta^{13}C_{SE\_50}$ curve. Next, we calculate the $\Delta$R value for each sample using the known-age approach in the online application *deltar* (Reimer and Reimer, 2017), based on the most recent Marine20 curve (Heaton et al., 2020). Finally, we calculate the weighted mean (Equation 2) and standard deviation of $\Delta$R following Reimer

and Reimer (2001), and provide a revised $\Delta$R estimate for the Gardar Drift. The uncertainty of $\Delta$R is determined as the maximum value of either the weighted uncertainty in the mean of $\Delta$R or the standard deviation of $\Delta$R, as in Equations 3 and 5. Our refined $\Delta$R estimate (-69±38 $^{14}$C yr) is similar to the value obtained from the marine reservoir database when 10 nearest points are used (-72±64 $^{14}$C yr)—although with better uncertainty estimates.

Weighted mean of $\Delta R = \mu = \dfrac{\sum_i \frac{\Delta R_i}{\sigma_i^2}}{\sum_i \frac{1}{\sigma_i^2}}$ ; *where $\sigma_i$ is the uncertainty in $\Delta R_i$* (Eq. 2)

Weighted uncertainty in mean of $\Delta R = \dfrac{1}{\sum_i \frac{1}{\sigma_i^2}}$ (Eq. 3)

Variance of $\Delta R = \dfrac{\frac{1}{n-1}\cdot\sum_i\left(\frac{\Delta R_i - \mu}{\sigma_i}\right)^2}{\frac{1}{n}\cdot\sum_i\frac{1}{\sigma_i^2}}$ (Eq. 4)

Standard Deviation of $\Delta R = \sqrt{variance}$ (Eq. 5)

**Table 2.** Revised ΔR estimate for Gardar Drift. "Known-ages" are derived from the [13]C Suess effect comparison, as shown in Figure 4. Weighted mean and standard deviation of ΔR is calculated following the method outlined in Calib, using Equations 2-5 (Reimer and Reimer, 2017; Reimer and Reimer, 2001).

| Core | Lab Code | Depth (cm) | [14]C age | ±1 σ | "Known-age" | ΔR (95% CI) |
|---|---|---|---|---|---|---|
| GS06-144-09MC-D | BE-19497.1.1 | 2.5 | 526 | 29 | 1918 | -79 ± 58 |
| GS06-144-09MC-D | BE-19498.1.1 | 4 | 565 | 29 | 1883 | -66 ± 58 |
| GS06-144-09MC-D | BE-19499.av | 5.5 | 603 | 48 | 1847 | -52 ± 96 |
| | | | | | **Weighted mean of ΔR =** | **-69 ± 38** |

**3.4 Revised age model for GS06-144-09 MC**

We use Bacon (version 2.5.0), the age-depth modelling approach that uses Bayesian statistics (Blaauw and Christen, 2011), operated through R (version 4.0.3)—a free software for statistical computing and graphics. A total of 10 [14]C AMS dates (Table 1) are calibrated through Bacon, using the most recent Marine20 curve (Heaton et al., 2020) and a ΔR value of -69±38 (this study) –assuming a constant ΔR value throughout the core. Since our ΔR estimate is based on the comparison with the [13]C Suess effect curve, we can only calculate a ΔR value for the last ~200 years with this approach. Although we assume relatively stable conditions over the last millennium (e.g., compared to glacial/interglacial changes), changes in ocean circulation and ventilation before this period will also effect the ΔR in the region (e.g., during the Little Ice Age, Spooner et al., 2020).

Additional tie points for 0.25 cm (1972 AD) and 7.5 cm (1800 AD) are used, based on the information obtained from the Suess effect curve. Based on the core top (0 cm) [14]C AMS date (>1950 AD) and the year the core was retrieved (2006 AD) the core top age should be between ~1950 and 2006 AD. As the core top age cannot be younger than 2006 AD, we use this information as a prior in Bacon to set a minimum age limit for the core top. According to the revised age model, the date for the core top (0- 0.5 cm) is 1977 AD. The average uncertainty for the last ~200 years (i.e., the 0-7.5 cm interval) is ~±42 years, and for the whole core (i.e., the 0-44 cm interval) is ±90 years. The resulting age depth plot is provided in Figure 5a. Although Bacon

selects the best age-depth model (i.e., red dotted lines in Figure 5), considering the sedimentation rate profile based on the prior information, the tie points at the core top and 1800 AD play a crucial role, providing a basis for sedimentation rates. This is also seen from Figure 5a, illustrated by the large range of $^{14}$C AMS dates that exceeds the calibration range of Marine20 due to bomb-carbon. This further underscores the need for independent chronological approaches particularly for the last century.

As a comparison, we also include the "known" calendar ages for samples 2.5 cm, 4 cm and 5.5 cm that were derived from the $\delta^{13}C_{SE\_50}$ comparison, together with their uncalibrated $^{14}$C dates in the Bacon input file. For all the tie points derived from the $\delta^{13}C_{SE\_50}$ comparisons we add a ±3-year uncertainty. Including the "known" calendar ages does not change the overall age model; but as expected, highly decreases the age model uncertainties for the last ~200 years (Figure 5b). Based on this, the core top age (0- 0.5 cm) is again 1977 AD. The average age model uncertainty for the last ~200 years (i.e., the 0-7.5 cm interval) is ±17.5 years. Below this point, the uncertainty increases (Average of ±84 years for the 0-44 cm interval) and is highly dependent on the uncertainty of the $^{14}$C AMS dates. The average sedimentation rates for the top 0-7.5 cm interval is 43 cm/kyr and 63 cm/kyr for the 7.5-44 cm interval. The average sedimentation rate of the core (0-44 cm) is 59 cm/kyr, giving a sample spacing of ~8.5 years per 0.5 cm sample.

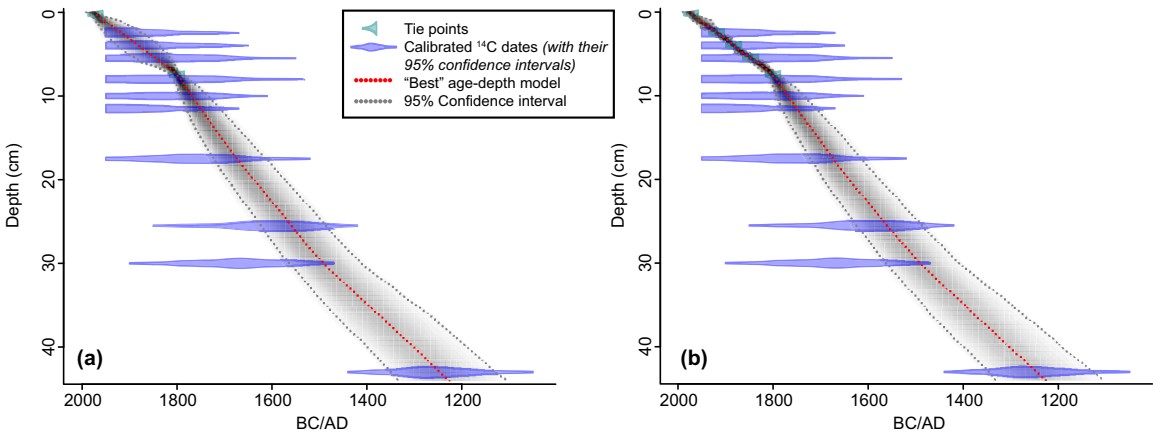

**Figure 5.** Age-depth plots of GS06-144-09 MC; (a) when additional tie points for 0.25 cm (1972 AD) and 7.5 cm (1800 AD) are used; and (b) when "known" calendar ages for samples 2.5 cm, 4 cm and 5.5 cm that were derived from the $\delta^{13}C_{SE\_50}$ comparison are used as additional tie points

### 3.5 SCP analysis

To cross check the validity of our Suess effect-derived age model, here we use another independent approach: Spheroidal carbonaceous fly ash particles (SCPs).

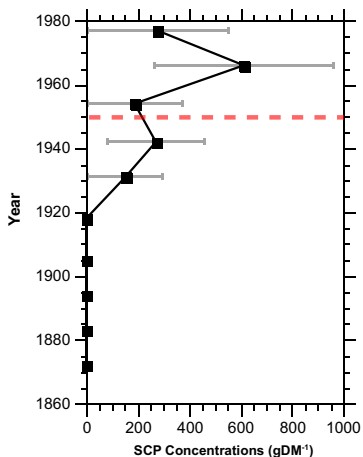

**Figure 6.** SCP Concentration profile of GS06-144-09 MC plotted vs the revised age model (as shown in Figure 5b). Dashed red line marks 1950.

SCP concentrations at GS06-144-09 MC are generally very low, varying between 152 and 616 gDM$^{-1}$. Based on our revised age model, SCP concentrations start to gradually increase during 1930s. A more marked increase in SCP concentrations occurs after 1954 and reach peak values at 1966, followed by a decline towards the core top. Supplementary Figure 4 shows a comparison of the GS06-144-09 MC SCP concentrations with previously published SCP profiles from the Apavatn Lake, Iceland (Rose, 2015) and Nunatak Lake, Greenland (Bindler et al., 2001; Rose, 2015). Despite similarly low concentrations,

both lake records show the same increase after ca. 1950, as the Gardar Drift marine sediment core. This suggests that the SCP concentrations at Gardar Drift follows a similar temporal pattern to the lake sediments in the region. Although the low SCP concentrations at GS06-144-09 MC result in considerable uncertainty for the SCP profile, the rapid increase after 1950s at GS06-144-09 MC is consistent with the SCP trend in the region and is consistent with our Suess effect-based revised age model.


**4 Discussion**

Our case study off Gardar Drift demonstrates the utility of two novel chronostratigraphic approaches that use anthropogenic signals (i.e., the oceanic $^{13}$C Suess effect change and SCP concentrations) in reducing age model uncertainties of recent high-

resolution marine archives. In addition, using a combination of $^{14}$C AMS dates and oceanic $^{13}$C Suess effect estimates, we further provide refined regional $^{14}$C reservoir corrections and uncertainties for Gardar Drift. Despite the similarity of our refined ΔR estimate to those available in the marine reservoir database (Reimer and Reimer, 2001), it is also important to note the shortcomings of our approach. For instance, by reading the corresponding "known-ages" from the $^{13}$C Suess effect curve to calculate ΔR, our approach assumes constant sedimentation rates, no bioturbation or reworking at the core top. Although

we do not see any visible traces of bioturbation in our core, we acknowledge that this is often not the case, and we rarely have sites with true "known-ages". One exception to this, and a potential to overcome this limitation, would be to use absolute age markers derived by identifying tephra layers and fingerprinting these to known volcanic eruptions. Yet this method is also only applicable in specific geologic settings and can also be affected by bioturbation– a limitation shared by all dating methods.

Bioturbation is one of the main sources of uncertainties of our approach as it will typically influence the age distributions and
smooth the record. Generally, the smoothing, or attenuation, is greater when the sedimentation rates are low (~10 cm/kyr) (Anderson, 2001). For instance, according to Anderson (2001) minimum attenuation (i.e., <5%) is observed only when sedimentation rates exceed 50-70 cm/kyr – a range often observed at sedimentary drift sites, such as the Gardar Drift. Given the average sedimentation rate of ~43 cm/kyr for the top 0 – 7.5 cm interval of our core (i.e., spanning an interval from ca. 1977 to 1800 AD), and sampling resolution of 0.5 cm, our ultimate chronological precision potentially achievable using these
methods would be ~12 years.

We further compare our revised age model based on anthropogenic signals with the previously published age model for Site GS06-144-09 MC-D. Figure 7 shows the $^{210}$Pb dates (Mjell et al., 2016), $^{14}$C dates and the information provided by the anthropogenic signals (i.e., $^{13}$C Suess effect derived tie points and the interval where the SCPs are present). The significant mismatch between the $^{210}$Pb and $^{14}$C dates once again highlights the need for independent approaches, as well as the potential
of using anthropogenic signals to improve age model constraints over the last two centuries.

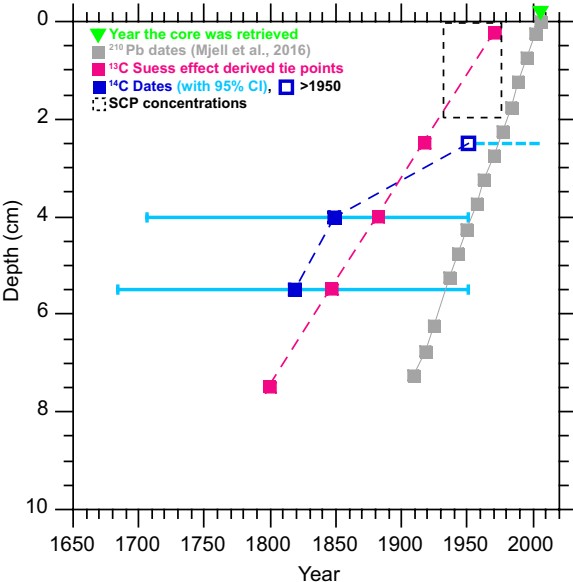

**Figure 7.** Age-depth plot for the top 10 cm of GS06-144-09 MC to highlight the differences (e.g., in sedimentation rate) between the original $^{210}$Pb based chronology (Mjell et al., 2016) vs the tie points derived based on anthropogenic signals (this study). $^{14}$C dates are calibrated
using Calib (version 8.2) (Stuiver and Reimer, 1993), using Marine20 and ΔR =-69±38 $^{14}$C yr.

One of the main differences between our revised age model and that of Mjell et al. (2016) is the core top age (1977 vs 2006, respectively). This once more emphasizes the need to validate [210]Pb based chronologies as well as the common assumption of the year a sediment core was retrieved as the core top age. Here we suggest and assume that the significant decline in our foraminiferal $\delta^{13}$C records over the last century is mainly caused by the oceanic [13]C Suess effect. This is particularly the case for our *G. bulloides* $\delta^{13}$C, where the actual decline in foraminiferal $\delta^{13}$C is the same as the [13]C Suess effect decline at 50 m depth. However, this may also be registered differently in other species. It is important to note that, although difficult to distinguish, our foraminiferal $\delta^{13}$C signals are also subject to natural climate variability. For instance, there are significant changes in the subpolar gyre circulation over the 20th century, and more specifically the observed productivity decline in the region (Spooner et al., 2020), will also be registered by our foraminiferal $\delta^{13}$C. Here, we have focused on the relative difference between average *G. bulloides* $\delta^{13}$C values over the industrial period vs the core top (i.e., sharpest [13]C decline due to Suess effect), and demonstrated the potential utility of the [13]C Suess effect approach in recent marine sediment chronologies. However, further sensitivity studies are needed to distinguish the effects of natural vs anthropogenic climate variability on foraminiferal $\delta^{13}$C records.

The scale of the, ongoing, Suess effect is now starting to exceed the entire range of $\delta^{13}$C exhibited through most open ocean environments (Eide et al., 2017) and, as such, it should be a dominant feature in records able to resolve short timescales. Indeed, the lack of this signal in a core top record suggests that either that modern sediments were not recovered and/or that sedimentation rates/bioturbation may confound sub-centennial scale interpretation of foraminiferal isotope records at a given core site.

Finally, Figure 8 shows the Sortable Silt record of Mjell et al. (2016) on its original age model that is based on [210]Pb and two [14]C dates vs the revised age model (as shown in Figure 5b) for GS06-144-09 MC-D, plotted together with the AMV index (Gray et al., 2004), to illustrate how our proxy-based interpretations for the 20th century might change with revised marine sediment chronologies.

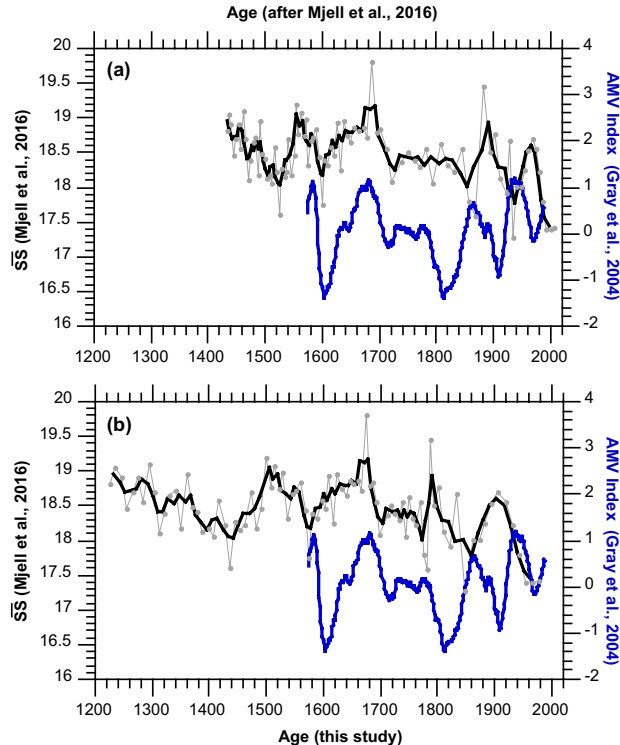

**Figure 8.** Sortable Silt mean grain size ($\overline{SS}$) as a proxy for Iceland-Scotland Overflow Water vigor (Mjell et al., 2016) vs AMV Index (Gray et al., 2004), plotted on (a) original age model (after Mjell et al., 2016) and (b) revised age model using anthropogenic signals (this study).

Although, marine based uncertainties over the last two centuries might still be too high (~±18 years in average) for a significant lead-lag comparison with observational records, our new approach based on anthropogenic signals provides an independent and valuable first step in refining age models and for validation of existing age model approaches and their assumptions.

**5 Summary and Conclusions**

High-resolution (i.e., decadal/multi-decadal) marine sediment records from the North Atlantic sedimentary drift sites are now emerging, with the potential to extend the instrumental records further back in time, distinguish natural climate variability vs anthropogenic and contextualize current changes. However, age model uncertainties, particularly over the 20[th] century pose major challenges, especially for integrating shorter instrumental records with these from extended marine archives. Recent sediments are dated using an array of methodologies, yet all have their own limitations (e.g., bomb-carbon, local reservoir

corrections for radiocarbon), either they are not applicable to all locations (e.g., tephrochronology) or can be below detection limits and requires another independent approach to confirm (e.g., $^{210}$Pb, $^{37}$Cs). Here we propose a new chronostratigraphic approach that uses anthropogenic signals to reduce age model uncertainties over the last two centuries. As a test application, we use the Gardar Drift sediment core GS06-144-09 MC and revise the age model at this site. Comparing planktonic $\delta^{13}$C

records of GS06-144-09 MC with oceanic [13]C Suess effect changes above the core location, we assign the beginning of the
industrial period (i.e., 1800 AD) in our core and similarly derive the core-top age. We further use a combination of [14]C AMS
dates and the [13]C Suess effect change estimates at our core location to calculate regional reservoir corrections at Gardar Drift.
Our refined ΔR estimate for Gardar Drift (-69±38 [14]C yr) is similar to the value obtained from the marine reservoir database
when 10 nearest points are used (-72±64 [14]C yr), however with better uncertainty estimates. Furthermore, to validate our [13]C
Suess effect-based age model we use another independent approach: Spheroidal carbonaceous fly ash particles (SCPs). The
rapid increase in SCP concentrations after 1950s at GS06-144-09 MC is consistent with the SCP trend in the region and our
[13]C Suess effect-based age model. Our new approach, based on anthropogenic signals, provides an independent and valuable
first step in refining age models and for validation of existing age model approaches and their assumptions.

### Data Availability

Data are available as supplementary information files.

### Author Contributions

N.I and U.S.N conceptualized the study. N.I refined the new age model approach together with U.S.N, F.C. and A.O. T.L.M
processed the multicore samples and performed stable isotope analysis. N.I processed samples for SCP analysis and conducted
SCP analysis together with N.L.R and D.J.R.T. U.S.N led the efforts on stable isotope analysis. A.O led the efforts on oceanic
[13]C Suess effect estimates for Gardar Drift. N.I led the writing effort and coordinated input from all co-authors.

### Competing Interests

The authors declare no competing interests.

### Acknowledgements

We thank the crew of R/V G. O. Sars, Institute of Marine Research (IMR), University of Bergen and the scientific party of
UiB Cruise No. GS06-144. We thank Marie Eide for her help with calculating oceanic [13]C Suess Effect estimates for our core
location at Gardar Drift. This study was funded by the Bjerknes Centre for Climate Research (BCCR) - Centre of Climate
Dynamics (SKD) Strategic Project PARCIM (Proxy Assimilation for Reconstructing Climate and Improving Model). Nil Irvalı
acknowledges additional support from the University of Bergen Meltzer Research Fund. Stable Isotope data were produced at
the Facility for Advanced Isotopic Research and Monitoring of Weather, Climate and Biogeochemical Cycling (FARLAB;
Research Council of Norway Grant: 245907).

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
