# Peer review of "Revising chronological uncertainties in marine archives using global anthropogenic signals: a case study on oceanic 13C Suess effect"

_EGUsphere, 2023_

## Author Comment (AC2)

**Response to Reviewer 2 Comments (RC2):**

Irvali and co-authors present a very interesting study with new ideas for establishing improved chronologies of modern sediments. Such efforts are much needed in paleoceanography because better chronologies for the upper, most recent sediments will allow the comparison of proxy data with instrumental time series and can therefore contribute to improving proxy calibrations and reconstructions.

The new data presented in this study include measurements of stable isotopes on foraminifera to detect the anthropogenic Suess effect and the occurrence of SCPs as additional age controls for the last few centuries.

Although the research idea and the data produced are of high quality, I have some major concerns about the chosen methods, the reporting of the data and structuring of the manuscript.

**We appreciate the reviewer's thoughtful feedback and suggestions. We address the specific comments below.**

In the introduction, the authors list various available techniques for dating recent marine sediments, and state that all of them have their own limitations and uncertainties. With so much data available for the studied sediment core, the ideal approach would thus be to combine all information into an integrated optimized age model for the last few centuries. However, the authors choose to create a core chronology based only on stable carbon isotopes measured on one species of foraminifera and their correlation to a model (based on the Suess effect), complemented by radiocarbon dating. The radiocarbon dating uses a reservoir age which is obtained from the abovementioned Suess correlation, so this is not independent either. The available Pb-210 and Cs-137 information is discarded and only included in the discussion, and the new SCP data is not actually used to build the age model but only to confirm the findings based on the isotopes.

Ultimately we agree, this manuscript is a case study useful for moving us in that direction, and specifically exploring the utility and consistency of using 13C-Suess. Once the utility and limitations are well defined, then a more integrative approach is certainly merited, although much of the information available is qualitative to semi-quantitative in nature.

On lines 108-109, the authors argue that the Cs-137 concentrations below 4 cm core depth were too low to detect and therefore these data were not used. One could argue that this is a result in itself; the anthropogenic Cs-137 isotope only occurs above 4 cm. I would consider this valuable information for the age model.

Thanks for this suggestion. We will fix and clarify this in the revised MS. The results from the Pb-210 and Cs-137 measurements from Mjell et al., 2016 *(their Supplementary Figure S1)* are shown below. In the case for core GS06-144-09MC, we think that Cs-137 is difficult to use unambiguously. Cs-137 is not only present above 4 cm, but also episodically below this depth (see below). As a result, it is not clear what this represents in chronological terms and so we consider it not useful to include in the age model.

[Figure]

**Figure 1.** Down-core variability of dry bulk density, unsupported $^{210}Pb$ content, and $^{137}Cs$ content in core GS06-144 09MC-D (Mjell et al., 2016).

The same is true for the radiocarbon age of the top of the core (Table 1, date KIA34242). The authors correctly state that this age indicates the presence of bomb carbon, and therefore younger than ~1957. This is another constraint that can be used to include in the age model. I don't understand why it is "not included in the age model". It is not possible to use it as a normal radiocarbon date, but an age constraint in the form of a maximum age should be possible to incorporate in the core chronology.

We thank the reviewer for bringing this up. This will be clarified in the revised MS. As the reviewer suggests, the core top radiocarbon date provides a valuable information and confirms that the core top is modern (0 cm >1957) (Line 235). However, here, we have instead used 2006 (the year the core was retrieved) to provide a maximum age and set a limit when building our age model in BACON, as the core top age (0 cm) cannot be younger than 2006.

The interpretation of the Suess effect in the foraminiferal isotope measurements is steered towards the final conclusions of the authors and does discuss other options. Only *bulloides*, and the combined stack are compared to the model output (Suppl. Table 1), but not the other individual species? Why not?

We prefer to use *G. bulloides* and the $\delta^{13}C$ stack (to cross-check the results we obtain from *G. bulloides*), instead of using all individual species (*N. incompta* and *G. inflata*). The main reason behind this is to look at the combined signal rather than habitat specific shifts and to avoid making further assumptions regarding foraminiferal habitat depths (e.g., particularly for *N. incompta*, which has a more variable depth habitat), and believe that using an average of 0-200 m would cover the habitat depths of all the planktonic foraminifers used in this work.

The curve fits are compared to various depth intervals from 12-0 to 5-0 cm (Lines 203-204), but why stop there? What about the curve fit coefficients at 4-0 cm, or even up to 1-0 cm?? The best fit is found for 7.5- 0 cm with r=0.73. Is this significant compared to the other intervals? For 0-5 cm, the r value is 0.69. Is that difference significant? And what would it be for 0-4 cm etc?

We appreciate this suggestion. We will extend the correlation analysis to the topmost part of the core in the revised MS and explore the possibility of including another test to evaluate the statistical differences between the curve fits for the various depth intervals.

After this simple statement: "7.5 cm must be 1800 AD", this is taken as fact, without any further discussion or even consideration of uncertainty. In SI Fig. 3 or in Fig. 4, one could imagine a very good fit between the red and blue curves if the red was more compressed and shifted to the right.

To make a more objective match of these two curves, and, instead of visual / wiggle matching, we have used polynomial curve fits and correlation analysis. Then, 7.5 cm is selected as it is based on the highest correlation coefficient between the Suess effect-*G. bulloides* comparison. Hence this statement is based on correlation analysis results presented in Supplementary Table 1.

The core top age is then determined as 1977, based on the findings above and a very simplified comparison between the upper parts of the isotope model data, and the *bulloides* measurements (Figure 3). This figure, instead, to me highlights the main difference between the data. One change is extremely abrupt, showing the major changes all in the top samples, while the other is gradual, with a decrease over 200 years. This difference is not sufficiently discussed. Also, this figure illustrates how the biggest change in *bulloides* is all in the top 1 cm of the core, and the moving average curve is here left out.

We will include a more detailed discussion of this. The moving average curve will also be included in the revised figure.

If the core top would actually be 1977, where is the rest of the sediment? Multi-cores are known to preserve the sediment-water interface, so what happened to the last 34 years?

The multicore GS06-144-09MC was retrieved in 2006, and if the core top is 1977, we would be 'missing' the last 29 years. Taking our average sedimentation rate for the 0-7.5 cm interval (~43 cm/kyr or 0.043 cm per year) at face value, we could then expect ~**1.2 cm** of sediment to be deposited in 29 years. However, the fact that we don't have a "modern" core top (or 1.2 cm more sediment) suggests changes in sedimentation rates, bottom flow speeds or bioturbation to be likely candidates.

From here onwards, this new age model is simply taken as the truth and no uncertainty is reported whatsoever. The term "known-age" is described between quotation marks, but that is not the same as discussing or reporting uncertainties. At some point (Line 345) a ±3-year uncertainty is reported, but it is not clear where this value comes from?

The authors claim to be able to deduce better estimates of the local ΔR values with lower uncertainties. They briefly acknowledge possible bioturbation, but don't discuss that the new ΔR values are based on the assumption that 0 cm is 1977 and 7.5 cm is 1800. The uncertainties derived from this should be included in the new reservoir age estimates.

We appreciate the reviewer' suggestions. We will state and clarify our assumptions in the revised MS.

Figure 7 shows a combination of some of the data, compared to the old Pb-210 age model. The very short discussion that follows just repeats how the new age model was made but fails to explain why the Pb-210 doesn't match. What could be reason for the mismatch? And would it not be possible to find a solution that satisfies all data? Perhaps the raw Pb-210 data could be reevaluated, and not just the resulting age-depth model.

We will include a more detailed discussion to explain this in the revised MS.

The manuscript often lacks a clear distinction between introduction, methods, results, interpretation or discussion. An example is paragraph 3.5 on SCP analyses. It combines most of the above in a single page.

Paragraph 3.5 will be rephrased to correct this in the revised MS.

In summary, the presented research idea is very promising, but in the current state, the methods and discussion do not make a convincing case that the new chronology is more reliable than the old one.

**Specific comments:**

- Lines 44 – 86. Several studies on recent marine sediments have used the increase of mercury concentrations as an anthropogenic marker for the last century. This could be added to your list. Example of a study from north of Iceland: https://doi.org/10.1371/journal.pone.0239373

We thank the reviewer for this suggestion. This will be included in the revised MS.

- Lines 116-117: *bulloides* was picked from the 250-300 µm size fraction, while *inflata* was picked from the 250-350 µm size fraction. Is this a typo, or did you include a 300 µm AND a 350 µm sieve?

This is correct. A narrower size range for *G. bulloides* is preferred as it is known to show variable stable isotope values based on differences in shell sizes. Hence, *G. bulloides* was only picked from the 250-300 µm fraction, while *G inflata* was picked from both 250-300 µm and 300-350 µm fractions.

- Line 155: "We set the starting point in time to 1800,…". How does the record look before that? Does it still look similar to the measured isotopes?

*G. bulloides* δ¹³C record vs the atmospheric δ¹³C from Rubino et al. (2013) is plotted below, spanning the last 1000 years . However, we avoid a direct comparison due to the sparsity of the data/sampling resolution before 1800.

[Figure]

**Figure 2.** *G. bulloides* δ¹³C record from Site GS06-144-09 MC-D *(blue, with 5-point mean)* plotted together with the atmospheric δ¹³C record of Rubino et al., 2013.

- Lines 191-193 does not give an objective description of the isotope results. Natural variability is described as "over the 10-44 cm core interval" which already includes the interpretation that thereafter, the changes are of anthropogenic origin. Why not explore the option that "natural variability goes to 4 cm? Or 1 cm depth? Why stop at 10?

We agree. The natural variability exists throughout the core (0-44 cm), and it would be impossible to distinguish this or state that there is no natural variability. In L191-193 we only suggest that there is a larger natural variability over the 10-44 cm interval of the core; while the anthropogenic influence increases and dominates the most recent interval/the core top. This has been discussed in more details in Lines 401-410. However, as also suggested by Reviewer 3, we will highlight more strongly the potential sensitivity of our approach to high natural variability in the revised MS and explore the possibility of including a sensitivity study on various cases.

- Lines 210-211: "suggesting 7.5 cm must be 1800 AD". This is too strong a statement. What about the uncertainty of this method and any critical discussion?

We will highlight the uncertainty of this approach in the revised MS.

- Lines 230-231: "gives us a rough estimate of which curve is most similar to our target curve (i.e., d13CSE_0-200), and overall agrees with our initial finding". Again, this is simply not critical enough. Would it also have agreed if you had other findings? Probably.

- Line 287: "the ΔR in the region is highly variable". What is meant here are the ΔR values based on the online database of calib.org, which are just a few observations. It is not the same as the actual ΔR values, so this should be clarified.

We agree. We will fix wording to clarify this in the revised MS.

- Line 323: Bioturbation is not limited to the top 10 cm of the core. Every depth level was once the top of the sediment, so bioturbation affects the entire core. The mixing or resulting smoothing of data is then over a ~10-cm window.

We will fix wording to clarify this in the revised MS.

---

## Author Comment (AC3)

**Response to Reviewer 3 Comments (RC3):**

This paper proposes a new chronostratigraphic approach that uses the oceanic $^{13}$C Suess effect and spheroidal carbonaceous particles (SCP) to improve the age models of marine sediment archives that cover the last few hundred to thousand years and could then serve to extend instrumental records back in time.

Extending instrumental records back in time is an important goal since it is the only way to improve our understanding of decadal to multidecadal climate variability and how this variability is currently affected by climate change. Marine sediments are one key climate archive in this respect but uncertainties associated with traditional dating techniques of marine sediment cores (e.g. $^{10}$Pb, $^{14}$C) are too large (up to ±30-50 y) to actually build continuous times series from sediment and instrumental records.

The paper is clearly written and its topic deserves publication. However, a number of points should be improved, as detailed below, before it can be accepted for publication.

**We appreciate the reviewer's thoughtful feedback and suggestions. We address the specific comments below.**

Main comments

- My first major comment is that there is a relatively large uncertainty with respect to the date of the starting point of the decline in atmospheric d$^{13}$C and with respect to the new core top age. But these uncertainties are both critical in the definition of the final revised age model. It is thus necessary to explore the impact of the uncertainty of these boundaries age on the final age model. This could be done by Monte Carlo or any other technique.

  Regarding the starting point of the decline in atmospheric d13C set at 1800, an error bar of ± 40 y seems reasonable given that the industrial revolution is dated between 1760 and 1840, depending on the authors. Note that this uncertainty does actually lead to an uncertainty in the definition of the core top age, which in turn is key in the definition of the new age model using the Bacon age-depth modeling software.

We thank the reviewer for this suggestion. We will explore the possibility of including a Monte Carlo (or similar) technique in the revised MS.

  In addition, l. 408, the authors discuss the possible impact of the changes in the subpolar gyre circulation over the 20$^{th}$century and productivity decline in this region on the *G. bulloides* d$^{13}$C. They write "we suspect the uncertainty based on natural climate variability to be minor in our core top age estimate". In my opinion, in a scientific article, it is required to go one step further than "suspecting". One way to go would be to carry out a sensitivity study to various school cases.

This is correct, the magnitude of natural variability will affect our core top age estimate. We will highlight more strongly the potential sensitivity of this approach to high natural variability in the revised MS and explore the possibility of including a sensitivity study on various cases.

- My second major comment concerns the estimate of the "known ages" for depths 2.5 cm, 4 cm and 5.5 cm by reading the corresponding ages from the d$^{13}$C$_{SE\_50}$ curve (Fig. 4): doing so the authors assume that the sedimentation rate is constant between 0 and 7.5 cm, which leads to a constant sedimentation rate over the top 7.5 cm of the core by construction

(as can be seen on Fig. 5b). This is a strong assumption which affects the final age model. It should be clearly described and its validity should be discussed.

The reviewer is correct. Our approach is based on, and assumes, constant sedimentation rates (and no bioturbation). We will include a more detailed discussion of this in the revised MS.

- A third important comment concerns the SCP profile : its resolution is too low to allow a verification of the age model obtained using the oceanic $^{13}$C Suess effect. The published SCP profiles of Rose (2015) seem to indicate that the SCP concentration peaks between 1970 and 1990 in regions adjacent to the core site, which does not match the SCP profile in the studied core plotted vs the revised age model in Fig. 6. This should be discussed. Also, an additional figure in the supplementary material showing the studied core SCP profile superimposed on published SCP profiles of Rose (2015) would be useful.

We appreciate this suggestion. We will include an additional supplementary figure and include a comparison of our Gardar Drift Core SCP Concentrations vs the published SCP profiles of Rose (2015). As also suggested by Reviewer 1, we will de-emphasize the SCP profile.

- 8 is practically not discussed. What does the revised age model suggest in terms of the relative phasing between the Iceland-Scotland overflow vigor and the Atlantic Multidecadal Variability?

Since the focus of this MS was more on the methodological aspect, we avoided including a detailed discussion on the relative phasing between ISOW – AMV, and instead aimed to highlight the need for refining uncertainties of, and improving, marine based age models for significant comparisons. However, we can include more discussion on this in the revised MS, taking our preliminary results (and uncertainties) at face value.

- Concerning the assessment of the ΔR to be applied to the radiocarbon dates: why do not the authors extract it from the GLODAP data set? This should provide a nice alternative to the CALIB marine reservoir database. It would be interesting to compare the ΔR currently computed by the authors with the ΔR based on the GLODAP data set. This would provide another estimate of the uncertainty associated with ΔR.

We thank the reviewer for pointing this out. In the revised MS, we will include a comparison with reservoir ages extracted from GLODAP.

- The conditions of validity of the assumption of Transient Steady State should be discussed. For instance, this assumption does not hold in case of changes in ocean circulation.

We agree. This will be qualified in the revised MS.

- Line 349-350: the average sedimentation rate over 0-44 cm is not really meaningful since it is larger below 30 cm than over the 7.5-30 cm depth interval, and larger below 7.5 cm than above 7.5 cm (Fig. 5).

The reviewer is correct. Sedimentation rates for different depth intervals (e.g., 0-7.5cm and 7.5-30 cm) will also be included in the revised MS.

**More minor comments:**

- 4 and 5 must contain a typo because they don't yield an uncertainty of ± 38 y for the weighted mean of ΔR, contrarily to what is indicated in Table 2.

This will be clarified in the revised MS. Equations 4 and 5 are correct. However, as stated in Line 305, the reported uncertainty of ΔR is determined as the highest value of either the standard deviation of ΔR (Eq. 5) or the weighted uncertainty in mean of ΔR (Eq. 3). In this case, the uncertainty of ± 38 yr (highest of the two values) is based on Eq. 3.

- 88: "that uses" should be replaced by "that use"

- To ease the reading, l. 181 should read "In the North Atlantic, *inflata* calcifies between 200 and 400 m south of 57°N, and between 100 and 200 m north of 57°N".

- 185-189: an additional figure of the d$^{13}$C stack together with the average $^{13}$C Suess effect change over 0-200 m would be useful in the supplementary material.

- 195: "even" seems unnecessary.

- 200: replace "found" by "computed" or "determined"

- 210: suppress "We show that,"

- 281: replace "would be" by "is"

- 301: replace "5 cm" by "5.5 cm"

- 378: replace "confirm" by "confirms"

- 7: the legend is too small

- 397-402: this is a repetition of what is written earlier in the article. Repetitions should be avoided.

- 412: replace "demonstrate" by "illustrate"

- 439: check the syntax.

We appreciate the reviewer for pointing out these. These will be corrected in the revised MS.

---

## Author Response (AR1)

**Author Response** for *"Revising chronological uncertainties in marine archives using global anthropogenic signals: a case study on oceanic $^{13}$C Suess effect"* (egusphere-2023-2845)

by Nil Irvalı, Ulysses S. Ninnemann, Are Olsen, Neil L. Rose, David J. R. Thornalley, Tor L. Mjell, François Counillon

**We thank the associate editor Richard Staff, and James Scourse and two anonymous reviewers for their positive feedback, thoughts, and suggestions on the manuscript. We address their specific comments below.**

**Response to Associate Editor Comments:**

The submission by Irvali and colleagues presents very interesting additional tools for the chronological modelling of marine sediment cores from the most recent centuries, and it is entirely appropriate for publication in Geochronology. As well as thanking the authors for this submission, I must also thank the three reviewers for their excellent thoughts on the manuscript and, obviously their time taken in consideration of this submission. The authors have also provided responses to these comments, which are absolutely to my satisfaction (pending my viewing of the revised manuscript!).

From my own perspective, I think that the reviewers have more than covered the thoughts that I had upon reading the manuscript in the first place, which included:

* Use of the term "ultra high resolution"; personally, I have worked a lot on varved lacustrine records and even in that context I have come across reviewers who object to that term! Ultimately, it is somewhat subjective, and there will always be an "even higher resolution" record somewhere (most probably under examination by whoever is reviewing your paper!)... But I do think that, in the current context, as noted by RC1, this term should be avoided.

This is a fair point. In the revised MS, we have now removed the "ultra high-resolution" term throughout and changed it to: "high-resolution".

* I agree with RC2 that, to obtain the "best possible" chronology, all lines of evidence should always be integrated... However, it is my opinion that the purpose of the present manuscript is NOT to derive the "best possible" chronology for THIS case study but, rather, to demonstrate the potential utility of the novel techniques, and therefore it is absolutely valid to use the techniques in more of a "standalone" manner, so as to independently test one another. Perhaps just a note to cover this might be worthwhile.

Thanks for this suggestion. We have now included a short paragraph in the methods section to cover this (Lines 149-154).

* More generally, whilst you are comparing chronological data obtained by different methods, it would obviously always be useful to know the "correct answer", at least at one or two places down your record, to be able to objectively know which method(s) are closest to the reality... But, obviously, we very rarely have any such sites with true "known ages". Tephra can provide such information, but only in specific geological settings... Could a note be incorporated to cover such thoughts?

We agree. We have now included this in the revised MS (Lines 513-515).

* As noted by both RC1 and RC2, sampling resolution MUST be an issue. Could some brief theoretical discussion of this be included, along the lines of "given an authentic sedimentation rate of x, and sampling resolution of y, our ultimate chronological precision potentially achievable using these methods is z"? Perhaps, noting the differences if each of these variables were tweaked, and/or the potential limiting factor of material availability.

Thanks for this suggestion. We have now included this in the Discussion section (Lines 520-523)

* A similar point is in relation to bioturbation (e.g., as discussed by both RC1 and RC2 and your responses to them): any bioturbation would serve to smooth your dataset, so can a brief theoretical note of how much such smoothing would limit the ultimate chronological resolution achievable by your methods?

We agree. Bioturbation is a potential caveat of our approach (an issue also raised by the reviewers). In the revised MS, we have now included a more detailed discussion on bioturbation (Lines 499-523).

* A concern that I share with RC1 and RC2 is in relation to the handling of the core top. Forgive my potential for misunderstanding your response to RC2... But it would be my understanding that the core top was BOTH constrained to be after AD 1957, but before 2006... Although "loose", I think that both of those limitations should be included in your model prior (and perhaps already are?). I would be absolutely wary of citing explanation that was reliant on something "different" happening JUST at the core top that hadn't been occurring throughout the broader time period studied (e.g., differences in bioturbation regime [discussed with RC1], differences in bottom flow speeds, differences in sedimentation rate [the latter two specifically noted in your response to RC2], or even the failure to collect the sediment/water interface...) without additional supporting information.

We agree. This is clarified in the revised MS (Lines 418-420), and in our response to RC2 (below).

* In your response to RC2, you note the "episodic" presence of 137Cs lower down the core; is this evidence of bioturbation and, if so, would it imply that that the latter is actually more significant than you are currently giving it credit for?

We have reworded to clarify this in the revised MS (Lines 141-147). However, it is correct that we cannot rule out bioturbation.

**Response to Reviewer 1 Comments (RC1):**

This paper reports a new approach to the establishment of reliable age models for very recent marine sediment cores (last 200 years). This is often a difficult and sometimes intractable problem for very recent cores and core-tops, because of the problems inherent in offsets between, for instance, radiocarbon, which has very poor precision for the last few hundred years, and age models based on Pb and Cs isotopes. This problem becomes significant because this is exactly the time period for which precise and accurate age models are required to calibrate proxy with instrumental data. These problems are rehearsed and explained well in this generally very well written contribution. In practice, most age models for recent marine sediments bring any data into play that might help construct and refine an age model and these data typically augment $^{14}C$ and Pb/Cs (notably tephra). This article – which is a case study - focuses on two additional approaches, using the oceanic $^{13}C$ Suess effect and spheroidal carbonaceous fly ash particles (SCP). The section on the Suess effect outlines an approach that will be of interest and use to many in the community, and the approach is novel; the section on SCP is included here as an accessory technique and is of less novelty. I have major concerns over the approach used to determine the age of the core-top.

**We thank Prof. James Scourse for his positive feedback and helpful suggestions. We address the specific comments below.**

**Specific comments**

1. I wonder whether the title should highlight/reflect the Suess effect approach? I think this is the most significant part of the paper and reference to this in the title would help flag this significance.

Thanks for this suggestion. In the revised MS, the title has been changed to highlight the Suess effect approach. Our new title is: *"Revising chronological uncertainties in marine archives using global anthropogenic signals: a case study on oceanic $^{13}C$ Suess effect"*.

2. Lines 47-48: the $^{14}C$ bomb-spike is introduced here as a confounding factor that increases uncertainties but it can provide a useful additional basis for assessing age if sufficient serial samples are available to define the spike.

We agree. This is now clarified in the revised MS (Lines 47-50).

3. Line 193: This sharp decline is only present in the final, single, 0.5 cm sample so the sampling resolution here could be problematic. Having higher resolution to define this decline more clearly would make the argument stronger. This issue is exacerbated considering the certain impact of bioturbation in the sediments, and the likely lateral variability in signals generated by bioturbation. Although in lines 321-322 the authors state that there were no visible signs of bioturbation, they acknowledge that bioturbation is common (actually ubiquitous) and that this will likely influence age distributions in the top 10 cm of the core. If there was no bioturbation the core would be laminated. I'm therefore more concerned with this somewhat over-interpreted approach to estimating core-top age than with the age-depth modelling. The authors should either consider strengthening this argument or deleting this section of the MS.

We fully appreciate and agree with the reviewer's comments on bioturbation. Strong bioturbation limits the application of our approach—a caveat we have now highlighted in more details in the revised MS (Lines 499-523).

4. Line 423: The term "ultra-high-resolution" should be reserved for archives that have annual to subannuual resolution.

This has now been changed to "high-resolution" in the revised MS.

**Response to Reviewer 2 Comments (RC2):**

Irvali and co-authors present a very interesting study with new ideas for establishing improved chronologies of modern sediments. Such efforts are much needed in paleoceanography because better chronologies for the upper, most recent sediments will allow the comparison of proxy data with instrumental time series and can therefore contribute to improving proxy calibrations and reconstructions.

The new data presented in this study include measurements of stable isotopes on foraminifera to detect the anthropogenic Suess effect and the occurrence of SCPs as additional age controls for the last few centuries.

Although the research idea and the data produced are of high quality, I have some major concerns about the chosen methods, the reporting of the data and structuring of the manuscript.

**We appreciate the reviewer's thoughtful feedback and suggestions. We address the specific comments below.**

In the introduction, the authors list various available techniques for dating recent marine sediments, and state that all of them have their own limitations and uncertainties. With so much data available for the studied sediment core, the ideal approach would thus be to combine all information into an integrated optimized age model for the last few centuries. However, the authors choose to create a core chronology based only on stable carbon isotopes measured on one species of foraminifera and their correlation to a model (based on the Suess effect), complemented by radiocarbon dating. The radiocarbon dating uses a reservoir age which is obtained from the abovementioned Suess correlation, so this is not independent either. The available Pb-210 and Cs-137 information is discarded and only included in the discussion, and the new SCP data is not actually used to build the age model but only to confirm the findings based on the isotopes.

Ultimately we agree, this manuscript is a case study useful for moving us in that direction, and specifically exploring the utility and consistency of using 13C-Suess. Once the utility and limitations are well defined, then a more integrative approach is certainly merited, although much of the information available is qualitative to semi-quantitative in nature.

On lines 108-109, the authors argue that the Cs-137 concentrations below 4 cm core depth were too low to detect and therefore these data were not used. One could argue that this is a result in itself; the anthropogenic Cs-137 isotope only occurs above 4 cm. I would consider this valuable information for the age model.

Thanks for this suggestion. We have now clarified this in the revised MS (Lines 141-147). The results from the Pb-210 and Cs-137 measurements from Mjell et al., 2016 *(their Supplementary Figure S1)* are shown below. In the case for core GS06-144-09MC, we think that Cs-137 is difficult to use unambiguously. Cs-137 is not only present above 4 cm, but trace amounts of Cs-137 is also episodically present below this depth (see below). As a result, it is

not clear what this represents in chronological terms and so we consider it not useful to include in the age model.

[Figure]

**Figure 1.** Down-core variability of dry bulk density, unsupported $^{210}$Pb content, and $^{137}$Cs content in core GS06-144 09MC-D (Mjell et al., 2016).

The same is true for the radiocarbon age of the top of the core (Table 1, date KIA34242). The authors correctly state that this age indicates the presence of bomb carbon, and therefore younger than ~1957. This is another constraint that can be used to include in the age model. I don't understand why it is "not included in the age model". It is not possible to use it as a normal radiocarbon date, but an age constraint in the form of a maximum age should be possible to incorporate in the core chronology.

We thank the reviewer for bringing this up. This is now clarified in the revised MS (Lines 418-420).  Based on the core top (0 cm) $^{14}$C AMS date (>1950 AD) and the year the core was retrieved (2006 AD) the core top age should be between ~1950 and 2006 AD. As the core top age cannot be younger than 2006 AD, we use this information as a prior in Bacon to set a minimum age limit for the core top.

The interpretation of the Suess effect in the foraminiferal isotope measurements is steered towards the final conclusions of the authors and does discuss other options. Only *bulloides*, and the combined stack are compared to the model output (Suppl. Table 1), but not the other individual species? Why not?

We prefer to use *G. bulloides* and the $\delta^{13}$C stack (to cross-check the results we obtain from *G. bulloides*), instead of using all individual species (*N. incompta* and *G. inflata*). The main reason behind this is to look at the combined signal rather than habitat specific shifts and to avoid making further assumptions regarding foraminiferal habitat depths (e.g., particularly for *N. incompta*, which has a more variable depth habitat), and believe that using an average of 0-200 m would cover the habitat depths of all the planktonic foraminifers used in this work.

The curve fits are compared to various depth intervals from 12-0 to 5-0 cm (Lines 203-204), but why stop there? What about the curve fit coefficients at 4-0 cm, or even up to 1-0 cm?? The best fit is found for 7.5- 0 cm with r=0.73. Is this significant compared to the other intervals? For 0-5 cm, the r value is 0.69. Is that difference significant? And what would it be for 0-4 cm etc?

*And from 'Specific Comments':* Lines 230-231: "gives us a rough estimate of which curve is most similar to our target curve (i.e., d13CSE_0-200), and overall agrees with our initial finding". Again, this is simply not critical enough. Would it also have agreed if you had other findings? Probably.

We appreciate this suggestion. We have now included a more in-depth discussion on this and evaluated the statistical differences between the correlation coefficients, e.g., for our best fit (r = 0.73) scenario and the second best fit (r=0.69) (Lines 267-272), and how this would change our core top age (Lines 319 -325).

After this simple statement: "7.5 cm must be 1800 AD", this is taken as fact, without any further discussion or even consideration of uncertainty. In SI Fig. 3 or in Fig. 4, one could imagine a very good fit between the red and blue curves if the red was more compressed and shifted to the right.

*And from 'Specific Comments':* Lines 210-211: "suggesting 7.5 cm must be 1800 AD". This is too strong a statement. What about the uncertainty of this method and any critical discussion?

To make a more objective match of these two curves, and, instead of visual / wiggle matching, we have used polynomial curve fits and correlation analysis. Then, 7.5 cm is selected as it is based on the highest correlation coefficient between the Suess effect-*G. bulloides* comparison.  However, we agree with the reviewer as placing the beginning of the Suess effect curve (1800 AD) on our G. bulloides d13C record is one of the challenges of our approach. In the revised MS, we have toned down this statement and included a more in-depth discussion on this. (Lines: 267-272; 319 -325)

The core top age is then determined as 1977, based on the findings above and a very simplified comparison between the upper parts of the isotope model data, and the *bulloides* measurements (Figure 3). This figure, instead, to me highlights the main difference between the data. One change is extremely abrupt, showing the major changes all in the top samples, while the other is gradual, with a decrease over 200 years. This difference is not sufficiently discussed. Also, this figure illustrates how the biggest change in *bulloides* is all in the top 1 cm of the core, and the moving average curve is here left out.

In the revised MS, we have included a more detailed discussion on the core top age (Lines: 319 -325), and Figure 3 is also revised to include the 5-point moving average curve.

If the core top would actually be 1977, where is the rest of the sediment? Multi-cores are known to preserve the sediment-water interface, so what happened to the last 34 years?

The top sample of the multicore is a 0.5 cm slice which means its average age, even if it captured some sediment from 2006 at the very surface, would be approximately the year 2000 (@0.25cm—the average of the sample slice).  Our Suess based estimate of 1977 is obviously older than this and implies at face value that nearly 1cm of sample was missing from the top of the core (given average sedimentation rates of ~43 cm/kyr or 0.043 cm per year). However, we cannot rule out that there is bioturbation which would incorporate older material up into this top sample biasing it slightly old or other factors such as small changes in local sedimentation patterns and sedimentation rates also at play.

From here onwards, this new age model is simply taken as the truth and no uncertainty is reported whatsoever. The term "known-age" is described between quotation marks, but that is

not the same as discussing or reporting uncertainties. At some point (Line 345) a ±3-year uncertainty is reported, but it is not clear where this value comes from?

The authors claim to be able to deduce better estimates of the local ΔR values with lower uncertainties. They briefly acknowledge possible bioturbation, but don't discuss that the new ΔR values are based on the assumption that 0 cm is 1977 and 7.5 cm is 1800. The uncertainties derived from this should be included in the new reservoir age estimates.

We appreciate the reviewer' suggestions. Our assumptions and uncertainties for our tie points are now clarified in the revised MS (Lines 319 -325).

Figure 7 shows a combination of some of the data, compared to the old Pb-210 age model. The very short discussion that follows just repeats how the new age model was made but fails to explain why the Pb-210 doesn't match. What could be reason for the mismatch? And would it not be possible to find a solution that satisfies all data? Perhaps the raw Pb-210 data could be reevaluated, and not just the resulting age-depth model.

We agree that, in general, an ideal approach would be to integrate all available information. However, our aim with this case study is to demonstrate the potential utility of two novel approaches. As also suggested by the editor, we have now included a short paragraph in the methods section to cover this (Lines 149-154).

The manuscript often lacks a clear distinction between introduction, methods, results, interpretation or discussion. An example is paragraph 3.5 on SCP analyses. It combines most of the above in a single page.

This is now fixed in the revised MS.

In summary, the presented research idea is very promising, but in the current state, the methods and discussion do not make a convincing case that the new chronology is more reliable than the old one.

**Specific comments:**

- Lines 44 – 86. Several studies on recent marine sediments have used the increase of mercury concentrations as an anthropogenic marker for the last century. This could be added to your list. Example of a study from north of Iceland: https://doi.org/10.1371/journal.pone.0239373

Thanks for this suggestion. We have now included this in the Introduction (Lines 53-55).

- Lines 116-117: *bulloides* was picked from the 250-300 μm size fraction, while *inflata* was picked from the 250-350 μm size fraction. Is this a typo, or did you include a 300 μm AND a 350 μm sieve?

This is correct. A narrower size range for *G. bulloides* is preferred as it is known to show variable stable isotope values based on differences in shell sizes. Hence, *G. bulloides* was only picked from the 250-300 μm fraction, while *G inflata* was picked from both 250-300 μm and 300-350 μm fractions.

- Line 155: "We set the starting point in time to 1800,…". How does the record look before that? Does it still look similar to the measured isotopes?

*G. bulloides* $\delta^{13}$C record vs the atmospheric $\delta^{13}$C from Rubino et al. (2013) is plotted below, spanning the last 1000 years . However, we avoid a direct comparison due to the sparsity of the data/sampling resolution before 1800.

[Figure]

**Figure 2.** *G. bulloides* $\delta^{13}$C record from Site GS06-144-09 MC-D *(blue, with 5-point mean)* plotted together with the atmospheric $\delta^{13}$C record of Rubino et al., 2013.

- Lines 191-193 does not give an objective description of the isotope results. Natural variability is described as "over the 10-44 cm core interval" which already includes the interpretation that thereafter, the changes are of anthropogenic origin. Why not explore the option that "natural variability goes to 4 cm? Or 1 cm depth? Why stop at 10?

We agree. The natural variability exists throughout the core (0-44 cm), and it would be impossible to distinguish this or state that there is no natural variability. In L240-244 we only suggest that there is a larger natural variability over the 10-44 cm interval of the core; while the anthropogenic influence increases and dominates the most recent interval/the core top. This has been discussed in more details in Lines 541-551.

- Line 287: "the ΔR in the region is highly variable". What is meant here are the ΔR values based on the online database of calib.org, which are just a few observations. It is not the same as the actual ΔR values, so this should be clarified.

Thanks for pointing this out, we agree. This is now clarified in the revised MS (Line 359).

- Line 323: Bioturbation is not limited to the top 10 cm of the core. Every depth level was once the top of the sediment, so bioturbation affects the entire core. The mixing or resulting smoothing of data is then over a ~10-cm window.

Bioturbation is a potential caveat of our approach (an issue also raised by the editor and RC1). In the revised MS, we have now included a more detailed discussion on bioturbation (Lines 499-523).

**Response to Reviewer 3 Comments (RC3):**

This paper proposes a new chronostratigraphic approach that uses the oceanic $^{13}$C Suess effect and spheroidal carbonaceous particles (SCP) to improve the age models of marine sediment archives that cover the last few hundred to thousand years and could then serve to extend instrumental records back in time.

Extending instrumental records back in time is an important goal since it is the only way to improve our understanding of decadal to multidecadal climate variability and how this variability is currently affected by climate change. Marine sediments are one key climate archive in this respect but uncertainties associated with traditional dating techniques of marine sediment cores (e.g. $^{10}$Pb, $^{14}$C) are too large (up to ±30-50 y) to actually build continuous times series from sediment and instrumental records.

The paper is clearly written and its topic deserves publication. However, a number of points should be improved, as detailed below, before it can be accepted for publication.

**We appreciate the reviewer's thoughtful feedback and suggestions. We address the specific comments below.**

Main comments

- My first major comment is that there is a relatively large uncertainty with respect to the date of the starting point of the decline in atmospheric d$^{13}$C and with respect to the new core top age. But these uncertainties are both critical in the definition of the final revised age model. It is thus necessary to explore the impact of the uncertainty of these boundaries age on the final age model. This could be done by Monte Carlo or any other technique.

    Regarding the starting point of the decline in atmospheric d13C set at 1800, an error bar of ± 40 y seems reasonable given that the industrial revolution is dated between 1760 and 1840, depending on the authors. Note that this uncertainty does actually lead to an uncertainty in the definition of the core top age, which in turn is key in the definition of the new age model using the Bacon age-depth modeling software.

We thank the reviewer for this suggestion. We acknowledge that one of the main challenges of our approach is to place the start of the 13C Suess effect decline (1800 AD) on our *G. bulloides* d13C record, although this seems to have little effect on our core top age (e.g., 3 years difference when 1800 AD is placed at 5 cm vs 7.5 cm).

    In addition, l. 408, the authors discuss the possible impact of the changes in the subpolar gyre circulation over the 20$^{th}$century and productivity decline in this region on the *G. bulloides* d$^{13}$C. They write "we suspect the uncertainty based on natural climate variability to be minor in our core top age estimate". In my opinion, in a scientific article, it is required to go one step further than "suspecting". One way to go would be to carry out a sensitivity study to various school cases.

This is correct, the magnitude of natural variability will affect our core top age estimate. We have now reworded this in the revised MS (Lines 547-557).

- My second major comment concerns the estimate of the "known ages" for depths 2.5 cm, 4 cm and 5.5 cm by reading the corresponding ages from the d$^{13}$C$_{SE\_50}$ curve (Fig. 4): doing so the authors assume that the sedimentation rate is constant between 0 and 7.5 cm, which leads to a constant sedimentation rate over the top 7.5 cm of the core by construction

(as can be seen on Fig. 5b). This is a strong assumption which affects the final age model. It should be clearly described and its validity should be discussed.

The reviewer is correct. Our approach is based on, and assumes, constant sedimentation rates (and no bioturbation). In the revised MS, we have now included a more detailed discussion on this (Lines 499-523).

- A third important comment concerns the SCP profile : its resolution is too low to allow a verification of the age model obtained using the oceanic $^{13}$C Suess effect. The published SCP profiles of Rose (2015) seem to indicate that the SCP concentration peaks between 1970 and 1990 in regions adjacent to the core site, which does not match the SCP profile in the studied core plotted vs the revised age model in Fig. 6. This should be discussed. Also, an additional figure in the supplementary material showing the studied core SCP profile superimposed on published SCP profiles of Rose (2015) would be useful.

We appreciate this suggestion. We have now included a comparison with the published SCP records from lakes that are closest to our core site, and show that the Gardar Drift SCP profile follows a similar temporal pattern to the lake sediments in the region (Supplementary Figure 4, Lines 487-495). As also suggested by RC2, we have also de-emphasized the SCP profile.

Regarding the comment on SCP Concentration peaks: the dates for SCP peaks vary on quite small geographical scales (Rose et al 1995; https://doi.org/10.1177/095968369500500308) so cannot easily be transferred to apply dates on the scales considered here with so few sites to compare with.

- 8 is practically not discussed. What does the revised age model suggest in terms of the relative phasing between the Iceland-Scotland overflow vigor and the Atlantic Multidecadal Variability?

Since the focus of this MS was more on the methodological aspect, we avoided including a detailed discussion on the relative phasing between ISOW – AMV, and instead aimed to highlight the need for refining uncertainties of, and improving, marine based age models for significant comparisons.

- Concerning the assessment of the ΔR to be applied to the radiocarbon dates: why do not the authors extract it from the GLODAP data set? This should provide a nice alternative to the CALIB marine reservoir database. It would be interesting to compare the ΔR currently computed by the authors with the ΔR based on the GLODAP data set. This would provide another estimate of the uncertainty associated with ΔR.

We thank the reviewer for pointing this out. In the revised MS we have now included a comparison with reservoir ages extracted from GLODAP (Lines 361-370).

- The conditions of validity of the assumption of Transient Steady State should be discussed. For instance, this assumption does not hold in case of changes in ocean circulation.

Here we follow Eide et al (2017) and the assumption of transient steady state (Gammon et al 1982, Tanhua et al., 2007) (Lines 187-191).

- Line 349-350: the average sedimentation rate over 0-44 cm is not really meaningful since it is larger below 30 cm than over the 7.5-30 cm depth interval, and larger below 7.5 cm than above 7.5 cm (Fig. 5).

The reviewer is correct. Sedimentation rates for different depth intervals (e.g., 0-7.5cm and 7.5-30 cm) are now included in the revised MS (Lines 444-445).

**More minor comments:**

- 4 and 5 must contain a typo because they don't yield an uncertainty of ± 38 y for the weighted mean of ΔR, contrarily to what is indicated in Table 2.

This is now clarified in the revised MS. Equations 4 and 5 are correct. However, as stated in Line 305, the reported uncertainty of ΔR is determined as the highest value of either the standard deviation of ΔR (Eq. 5) or the weighted uncertainty in mean of ΔR (Eq. 3). In this case, the uncertainty of ± 38 yr (highest of the two values) is based on Eq. 3.

- 88: "that uses" should be replaced by "that use"

- To ease the reading, l. 181 should read "In the North Atlantic, *inflata* calcifies between 200 and 400 m south of 57°N, and between 100 and 200 m north of 57°N".

- 185-189: an additional figure of the d[13]C stack together with the average [13]C Suess effect change over 0-200 m would be useful in the supplementary material.

- 195: "even" seems unnecessary.

- 200: replace "found" by "computed" or "determined"

- 210: suppress "We show that,"

- 281: replace "would be" by "is"

- 301: replace "5 cm" by "5.5 cm"

- 378: replace "confirm" by "confirms"

- 7: the legend is too small

- 397-402: this is a repetition of what is written earlier in the article. Repetitions should be avoided.

- 412: replace "demonstrate" by "illustrate"

- 439: check the syntax.

We appreciate the reviewer for pointing out these. These are corrected in the revised MS.